# Bellman Unbiasedness: Toward Provably Efficient Distributional Reinforcement Learning with General Value Function Approximation

Taehyun Cho[1]  Seungyub Han[1]  Seokhun Ju[1]  Dohyeong Kim[1]  Kyungjae Lee[2]  Jungwoo Lee[1]

## Abstract

Distributional reinforcement learning improves performance by capturing environmental stochasticity, but a comprehensive theoretical understanding of its effectiveness remains elusive. In addition, the intractable element of the infinite dimensionality of distributions has been overlooked. In this paper, we present a regret analysis of distributional reinforcement learning with general value function approximation in a finite episodic Markov decision process setting. We first introduce a key notion of *Bellman unbiasedness* which is essential for exactly learnable and provably efficient distributional updates in an online manner. Among all types of statistical functionals for representing infinite-dimensional return distributions, our theoretical results demonstrate that only moment functionals can exactly capture the statistical information. Secondly, we propose a provably efficient algorithm, `SF-LSVI`, that achieves a tight regret bound of $\tilde{O}(d_E H^{\frac{3}{2}} \sqrt{K})$ where $H$ is the horizon, $K$ is the number of episodes, and $d_E$ is the eluder dimension of a function class.

Distributional reinforcement learning (DistRL) (Bellemare et al., 2017; Rowland et al., 2019; Choi et al., 2019; Kim et al., 2024) is an advanced approach to reinforcement learning (RL) that focuses on the entire probability distribution of returns rather than solely on the expected return. By considering the full distribution of returns, distRL provides deeper insight into the uncertainty of each action, such as the mode or median. This framework enables us to make safer and more effective decisions that account for various risks (Chow et al., 2015; Son et al., 2021; Greenberg et al., 2022; Kim et al., 2023), particularly in complex real-world situations, such as robotic manipulation (Bodnar et al., 2019), neural response (Muller et al., 2024), stratospheric balloon navigation (Bellemare et al., 2020), algorithm discovery (Fawzi et al., 2022), and several game benchmarks (Bellemare et al., 2013; Machado et al., 2018). While the distributional approach offers richer information, two key theoretical challenges are introduced that distinguish it from expectation-based RL.

**Infinite-dimensionality of distribution.** In practice, distributions contain an infinite amount of information, and we must resort to approximations using a finite number of parameters or statistical functionals, such as categorical (Bellemare et al., 2017) and quantile representations (Dabney et al., 2018b). However, previous works often conducted analyses while overlooking these intractable nature of distributions. Additionally, not all statistical functionals can be *exactly learned* through the Bellman operator, as the meaning of statistical functionals is not preserved after updates. For example, the median is not preserved under the Bellman updates, as the median of a mixture of two distributions does not equal the mixture of their medians. Thus, a fundamental question arises:

> *"For a given statistical functional, does there exist a corresponding Bellman operator that ensures exactness?"*

To formalize this issue, Rowland et al. (2019) introduced *Bellman closedness*, which characterizes statistical functionals that can be exactly learned in the presence of a corresponding Bellman operator.

**Online distributional update.** In this paper, we focus on developing an algorithm that efficiently explores from a regret minimization perspective while simultaneously performing distributional Bellman updates in an online manner. One possible approach to addressing this problem is to first update the policy using an existing provably efficient non-distributional RL algorithm and then estimate the distribution via additional rollouts. However, decoupling these two processes introduces several drawbacks. First, adding extra rollouts solely for distribution estimation is sample-inefficient, and the limited number of rollouts inevitably introduce accumulated approximation errors in the estimated distribution throughout the learning process.

---

[1]Seoul National University, Seoul, South Korea [2]Korea University, Seoul, South Korea. Correspondence to: Kyungjae Lee <kyungjae_lee@korea.ac.kr>, Jungwoo Lee <junglee@snu.ac.kr>.

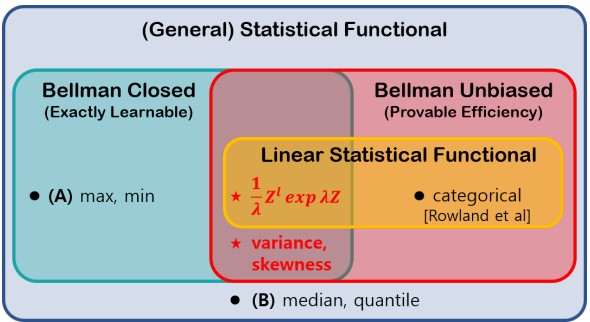

*Figure 1.* **Venn-Diagram of Statistical Functional Classes.** The diagram illustrates categories of statistical functional. **(Yellow ∩ Blue)** Within the linear statistical functional class, Rowland et al. (2019) showed that the only functionals satisfying Bellman closedness are moment functionals. **(Red ∩ Blue)** We extend this concept by introducing the notion of *Bellman unbiasedness*, which not only covers moment functionals but also includes central moment functionals from the broader class including nonlinear statistical functionals. **(Yellow ∩ Blue$^c$)** According to Lemmas 3.2 and 4.4 of Rowland et al. (2019), categorical functionals are linear but not Bellman closed. **(A)** Maximum and minimum functionals are Bellman closed, while they are not unbiasedly estimable. **(B)** Median and quantile functionals are neither Bellman closed nor unbiased, highlighting that they are not proper to encode the distribution in terms of exactness. The proofs corresponding to each region are provided in Appendix C.

Moreover, the estimation is confined to the return distribution of the executed policy, making it difficult to reuse for estimating distributions under different policies, thereby moving further away from off-policyness.

To overcome those two fundamental challenges inherent to DistRL, we take a closer look at the distributional Bellman update and revisit what additional properties of statistical functionals, beyond Bellman closedness, are required to construct online DistRL algorithms that are not only exactly learnable but also *provably efficient* in terms of regret. In this context, we identify the following additional issues that arise when using statistical functionals for updates instead of the full distribution:

- Representing a mixture distribution with a finite, fixed number of parameters leads to approximation errors during the update. For example, when expressing the mixture of two distributions, each represented by $N$ parameters, compressing $2N$ into $N$ parameters in the mixture results in inevitable information loss.

- Due to the unknown nature of the transition $\mathbb{P}(\cdot|s,a)$, the target distribution is estimated by sampling the next state $s'$. Hence, the statistical functionals of the target distribution should be unbiasedly estimated using the statistical functionals from the sampled distribution.

In this paper, we introduce a key concept, *Bellman unbiasedness*, for precise information learnability of a distribution from a finite number of samples in an online setting. As shown in Figure 1, we prove that the moment functional remains the only solution in a class that includes *nonlinear* statistical functionals that satisfies both properties. We then discuss the inherent intractability of *distributional Bellman completeness* (distBC) – a structural assumption previously

defined in the literature (Wang et al., 2023; Chen et al., 2024) – and investigate the benefits of redesigning this concept using a collection of statistical functionals. Finally, we propose a provably efficient statistical functional RL algorithm with general value function approximation, called `SF-LSVI`.

In summary, our main contributions are as follows:

- Introduce a key property of Bellman unbiasedness for exactly learnable and provably efficient online distRL algorithm. We show that the moment functional is the unique structure in a class including nonlinear statistical functionals.

- Describe the inherent intractability of infinite-dimensional distributions and analyze how hidden approximation error prevents the design of provably efficient algorithms. To address this, we revisit the existing structural assumption of distributional Bellman Completeness through a statistical functional lens.

- Propose exactly learnable and provably efficient distRL algorithm called `SF-LSVI`, achieving a tight regret upper bound $\tilde{O}(d_E H^{\frac{3}{2}}\sqrt{K})$. [1] Our framework yields a tighter regret bound with a weaker structural assumption compared to prior results in distRL.

# 1. Related Work

---

[1] We ignore poly-log terms in $H, S, A, K$ in the $\tilde{O}(\cdot)$ notation.

[2] In Chen et al. (2024), the regret bound is written as $\tilde{O}(d_E L_\infty(\rho) H \sqrt{K})$, where $L_\infty(\rho)$ represents the lipschitz constant of the risk measure $\rho$, i.e., $|\rho(Z) - \rho(Z')| \le L_\infty(\rho)\|F_Z - F_{Z'}\|_\infty$. Since $L_\infty(\rho) \ge H$ in risk-neutral setting, we translate the regret bound into $\tilde{O}(d_E H^2 \sqrt{K})$.

*Table 1.* Comparison for different methods under distributional RL framework. $\mathcal{H}$ represents a subspace of infinite-dimensional space $\mathcal{F}^\infty$. To bound the eluder dimesion $d_E$, Wang et al. (2023) and Chen et al. (2024) assumed the discretized reward MDP.

| Algorithm | Regret | Eluder dimension $d_E$ | Bellman Completeness | MDP assumption | Finite Representation | Exactly Learnable |
|---|---|---|---|---|---|---|
| O-DISCO (Wang et al., 2023) | $\tilde{\mathcal{O}}(\text{poly}(d_E H)\sqrt{K})$ | $\dim_E(\mathcal{H}, \epsilon)$ | distributional BC | discretized reward, small-loss bound | ✗ | ✗ |
| V-EST-LSR (Chen et al., 2024) | $\tilde{\mathcal{O}}(d_E H^2 \sqrt{K})$ [2] | $\dim_E(\mathcal{H}, \epsilon)$ | distributional BC | discretized reward, lipschitz continuity | ✗ | ✗ |
| SF-LSVI [Ours] | $\tilde{\mathcal{O}}(d_E H^{\frac{3}{2}} \sqrt{K})$ | $\dim_E(\mathcal{F}^N, \epsilon)$ | statistical functional BC | none | ✓ | ✓ |

**Distributional RL.** In classical RL, the Bellman equation, which is based on expected returns, has a closed-form expression. However, it remains unclear whether any statistical functionals of return distribution always have their corresponding closed-form expressions. Rowland et al. (2019) introduced the notion of *Bellman closedness* for collections of statistical functionals that can be updated in a closed form via Bellman update. They showed that the only Bellman-closed statistical functionals in the discounted setting are the moments $\mathbb{E}_{Z \sim \eta}[Z^k]$. More recently, Marthe et al. (2023) proposed a general framework for distRL, where the agent plans to maximize its own utility functionals instead of expected return, formalizing this property as *Bellman Optimizability*. They further demonstrated that in the undiscounted setting, the only $W_1$-continuous and linear Bellman optimizable statistical functionals are exponential utilities $\frac{1}{\lambda} \log \mathbb{E}_{Z \sim \eta}[\exp(\lambda Z)]$.

In practice, C51 (Bellemare et al., 2017) and QR-DQN (Dabney et al., 2018b) are notable distributional RL algorithms where the convergence guarantees of sampled-based algorithms are proved (Rowland et al., 2018; 2023). Dabney et al. (2018a) expanded the class of policies on arbitrary distortion risk measures by taking the based distribution non-uniformly and improve the sample efficiency from their implicit representation of the return distribution. Cho et al. (2023) highlighted the drawbacks of optimistic exploration in distRL, introducing a randomized exploration that perturbs the distribution when the agent selects next action.

**RL with General Value Function Approximation.** Regret bounds have been studied for a long time in online RL, across various domains such as bandit (Lattimore & Szepesvári, 2020; Abbasi-Yadkori et al., 2011; Russo & Van Roy, 2013), tabular RL (Kakade, 2003; Auer et al., 2008; Osband & Van Roy, 2016; Osband et al., 2019; Jin et al., 2018), and linear function approximation (Jin et al., 2020; Wang et al., 2019; Zanette et al., 2020). In recent years, deep RL has shown significant performance using deep neural networks as function approximators, and attempts have been made to analyze whether it is efficient in terms of general function approximation (Jin et al., 2021; Agarwal et al., 2023). Wang et al. (2020) established a

provably efficient RL algorithm with general value function approximation based on the eluder dimension $d_E$ (Russo & Van Roy, 2013) and achieves a regret upper bound of $\tilde{O}(\text{poly}(d_E H)\sqrt{K})$. To circumvent the intractability from computing the upper confidence bound, Ishfaq et al. (2021) injected the stochasticity on the training data and get the optimistic value function instead of upper confidence bound, enhancing computationally efficiency. Beyond risk-neutral setting, several prior works have shown regret bounds under risk-sensitive objectives (e.g., entropic risk (Fei et al., 2021; Liang & Luo, 2022), CVaR (Bastani et al., 2022)), which align with our approach in that they are built on a distribution framework. Liang & Luo (2022) achieved the regret upper bound of $\tilde{O}(\exp(H)\sqrt{|\mathcal{S}|^2|\mathcal{A}|H^2K})$ and the lower bound of $\Omega(\exp(H)\sqrt{|\mathcal{S}||\mathcal{A}|HK})$ in tabular setting.

**DistRL with General Value Function Approximation.** Recently, only few efforts have aimed to bridge the gap between two fields. Wang et al. (2023) proposed a distributional RL algorithm, O-DISCO, which enjoys small-loss bound by using a log-likelihood objective. Similarly, Chen et al. (2024) provided a risk-sensitive RL framework with static lipschitz risk measure. While these studies analyze within a distributional framework, they do not address the intractability of implementation in infinite-dimensional space of distributions. In contrast, our approach focuses on a statistical functional framework, providing a detailed comparison with other distRL methods as shown in Table 1.

## 2. Preliminaries

**Episodic MDP.** We consider a episodic Markov decision process which is defined as a $\mathcal{M} = (\mathcal{S}, \mathcal{A}, H, \mathbb{P}, r)$ characterized by state space $\mathcal{S}$, action space $\mathcal{A}$, horizon length $H$, transition kernels $\mathbb{P} = \{\mathbb{P}_h\}_{h \in [H]}$, and reward $r = \{r_h\}_{h \in [H]}$ at step $h \in [H]$. The agent interacts with the environment across $K$ episodes. For each $k \in [K]$ and $h \in [H]$, $\mathbb{H}_h^k = (s_1^1, a_1^1, \ldots, s_H^1, a_H^1, \ldots, s_h^k, a_h^k)$ represents the history up to step $h$ at episode $k$. We assume the reward is bounded by $[0, 1]$ and the agent always transit to terminal state $s_{\text{end}}$ at step $H + 1$ with $r_{H+1} = 0$.

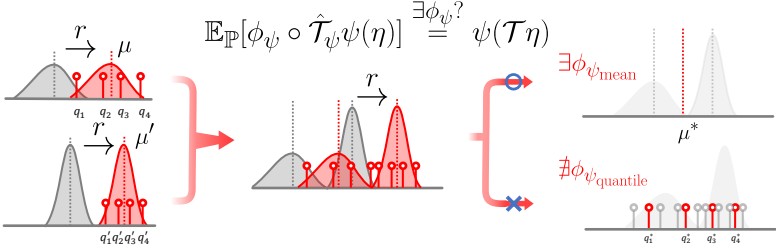

*Figure 2.* Illustrative representation of sketch-based Bellman updates for a mixture distribution. Instead of updating the distributions directly, each sampled distribution is embedded through a sketch $\psi$ (e.g., mean $\mu$, quantile $q_i$). The transformation $\phi_\psi$ aims to compress the mixture distribution into the same number of parameters, ensuring unbiasedness to prevent information loss.

**Policy and Value Functions.** A (deterministic) policy $\pi$ is a collection of $H$ functions $\{\pi_h : \mathcal{S} \to \mathcal{A}\}_{h=1}^H$. Given a policy $\pi$, a step $h \in [H]$, and a state-action pair $(s, a) \in \mathcal{S} \times \mathcal{A}$, the $Q$ and $V$-function are defined as $Q_h^\pi(s,a)(: \mathcal{S} \times \mathcal{A} \to \mathbb{R}) := \mathbb{E}_\pi \left[ \sum_{h'=h}^H r_{h'}(s_{h'}, a_{h'}) \mid s_h = s, a_h = a \right]$ and $V_h^\pi(s)(: \mathcal{S} \to \mathbb{R}) := \mathbb{E}_\pi \left[ \sum_{h'=h}^H r_{h'}(s_{h'}, a_{h'}) \mid s_h = s \right]$.

**Random Variables and Distributions.** For a sample space $\Omega$, we extend the definition of the $Q$-function into a random variable and its distribution,

$$Z_h^\pi(s,a)(: \mathcal{S} \times \mathcal{A} \times \Omega \to \mathbb{R})$$
$$:= \sum_{h'=h}^H r_{h'}(s_{h'}, a_{h'}) \mid s_h = s, a_h = a, a_{h'} = \pi_{h'}(s_{h'}),$$
$$\eta_h^\pi(s,a)(: \mathcal{S} \times \mathcal{A} \to \mathscr{P}(\mathbb{R})) := \mathrm{law}(Z_h^\pi(s,a)).$$

Analogously, we extend the definition of $V$-function by introducing a bar notation.

$$\bar{Z}_h^\pi(s)(: \mathcal{S} \times \Omega \to \mathbb{R})$$
$$:= \sum_{h'=h}^H r_{h'}(s_{h'}, a_{h'}) \mid s_h = s, a_{h'} = \pi_{h'}(s_{h'}),$$
$$\bar{\eta}_h^\pi(s)(: \mathcal{S} \to \mathscr{P}(\mathbb{R})) := \mathrm{law}(\bar{Z}_h^\pi(s)).$$

Note that $\bar{Z}_h^\pi(s) = Z_h^\pi(s, \pi(s))$ and $\bar{\eta}_h^\pi(s) = \eta_h^\pi(s, \pi(s))$. We use $\pi^\star$ to denote an optimal policy ( *i.e.*, $\pi_h^\star(\cdot|s) = \arg\max_\pi V_h^\pi(s)$ ) and denote $V_h^\star(s) = V_h^{\pi^\star}(s), Q_h^\star(s,a) = Q_h^{\pi^\star}(s,a), \eta_h^\star(s,a) = \eta_h^{\pi^\star}(s,a)$, and $\bar{\eta}_h^\star(s) = \bar{\eta}_h^{\pi^\star}(s)$. For notational simplicity, we denote the expectation over transition, $[\mathbb{P}_h V_{h+1}^\pi](s,a) = \mathbb{E}_{s' \sim \mathbb{P}_h(\cdot|s,a)} V_{h+1}^\pi(s')$, $[\mathbb{P}_h \bar{Z}_{h+1}^\pi](s,a) = \mathbb{E}_{s' \sim \mathbb{P}_h(\cdot|s,a)} \bar{Z}_{h+1}^\pi(s')$, and $[\mathbb{P}_h \bar{\eta}_{h+1}^\pi](s,a) = \mathbb{E}_{s' \sim \mathbb{P}_h(\cdot|s,a)} \bar{\eta}_{h+1}^\pi(s')$.[3] For brevity, we refer to $\bar{\eta}^\pi$ simply as $\bar{\eta}$.

In the episodic MDP, the agent aims to learn the optimal policy through a fixed number of interactions with the environment across a number of episodes. At the beginning of each episode $k(\in [K])$, the agent starts at the initial state

---

[3]Note that $\mathbb{E}_{s' \sim \mathbb{P}_h(\cdot|s,a)} \bar{\eta}_{h+1}^\pi(s')$ is a mixture distribution.

$s_1^k$ and choose a policy $\pi^k$. In step $h(\in [H])$, the agent observes $s_h^k(\in \mathcal{S})$, takes an action $a_h^k(\in \mathcal{A}) \sim \pi_h^k(\cdot|s_h^k)$, receives a reward $r_h(s_h^k, a_h^k)$, and the environment transits to the next state $s_{h+1}^k \sim \mathbb{P}_h(\cdot|s_h^k, a_h^k)$. Finally, we measure the suboptimality of an agent by its regret, which is the accumulated difference between the ground truth optimal and the return received from the interaction. The regret after $K$ episodes is defined as $\mathrm{Reg}(K) = \sum_{k=1}^K V_1^\star(s_1^k) - V_1^{\pi^k}(s_1^k)$.

**Distributional Bellman Optimality Equation.** Recall that $\eta_h^\star$ satisfies the following optimality equation:

$$\eta_h^\star(s,a) = (\mathcal{T}_h \eta_{h+1}^\star)(s,a)$$
$$:= \mathbb{E}_{s' \sim \mathbb{P}_h(\cdot|s,a), a' \sim \pi_h^\star(\cdot|s')}[(\mathcal{B}_{r_h})_\# \eta_{h+1}^\star(s', a')]$$
$$= (\mathcal{B}_{r_h})_\# [\mathbb{P}_h \eta_{h+1}^\star](s,a)$$

where $\mathcal{B}_r : \mathbb{R} \to \mathbb{R}$ is defined by $\mathcal{B}_r(x) = r+x$, and $g_\# \eta \in \mathscr{P}(\mathbb{R})$ is the pushforward of the distribution $\eta$ through $g$ (*i.e.*, $g_\# \eta(A) = \eta(g^{-1}(A))$ for any Borel set $A \subseteq \mathbb{R}$).

**Additional Notations.** For a given $N$, we denote an $N-$dimensional function class $\mathcal{F}^N := \mathcal{F}^{(1)} \times \cdots \times \mathcal{F}^{(N)} \subseteq \left\{ f = [f^{(1)}, \cdots, f^{(N)}] : \mathcal{S} \times \mathcal{A} \to \mathbb{R}^N \right\}$. Given a dataset $\mathcal{D} = \{(s_t, a_t, [z_t^{(1)}, \ldots, z_t^{(N)}])\}_{t=1}^{|\mathcal{D}|} \subseteq \mathcal{S} \times \mathcal{A} \times \mathbb{R}^N$, a set of state-action pairs $\mathcal{Z} = \{(s_t, a_t)\}_{t=1}^{|\mathcal{Z}|} \subseteq \mathcal{S} \times \mathcal{A}$ and for a function $f : \mathcal{S} \times \mathcal{A} \to \mathbb{R}^N$, we define the norm $\|f^{(n)}\|_\infty, \|f\|_{\infty,1}, \|f\|_\mathcal{D}, \|f\|_\mathcal{Z}$ as written in Appendix A. For a set of (vector-valued) functions $\mathcal{F}^N \subseteq \{f : \mathcal{S} \times \mathcal{A} \to \mathbb{R}^N\}$, the width function of $(s,a)$ is defined as $w^{(n)}(\mathcal{F}^N, s, a) := \max_{f,g \in \mathcal{F}^N} |f^{(n)}(s,a) - g^{(n)}(s,a)|$.

## 3. Statistical Functionals in Distributional RL

In this section, we define two key concepts in the distRL framework: the *statistical functional* and the *sketch*. We also illustrate *Bellman closedness*, a crucial property from Bellemare et al. (2023). Next, we introduce *Bellman unbiasedness*, a novel concept that complements the previous property and is essential for provable efficiency. As shown in Figure 2, quantile functionals cannot be updated in an

unbiased manner (as proved in Theorem 3.3), demonstrating that only certain sketches can be updated exactly. We then show that the only sketch satisfying both properties is the moment functional, which is unique among statistical functionals. Finally, we discuss the intractability of the previous structural assumption, distributional Bellman Completeness, and its tendency to cause linear regret. To address this, we introduce *statistical functional Bellman Completeness*, a relaxed assumption, and explain why it satisfies both properties.

### 3.1. Bellman Closedness

**Definition 3.1** (Statistical functionals, Sketch; (Bellemare et al., 2023)). A **statistical functional** is a mapping from a probability distribution to a real value $\psi : \mathscr{P}(\mathbb{R}) \to \mathbb{R}$. A **sketch** is a vector-valued function $\psi_{1:N} : \mathscr{P}(\mathbb{R}) \to \mathbb{R}^N$ specified by an $N$-tuple where each component is a statistical functional,

$$\psi_{1:N}(\cdot) = (\psi_1(\cdot), \cdots, \psi_N(\cdot)).$$

We denote the domain of sketch as $\mathscr{P}_{\psi_{1:N}}(\mathbb{R})$ and its image as $I_{\psi_{1:N}} = \{\psi_{1:N}(\bar{\eta}) : \bar{\eta} \in \mathscr{P}_{\psi_{1:N}}(\mathbb{R})\}$. We further extend to state return distribution functions $\psi_{1:N}(\bar{\eta}) = \left(\psi_{1:N}(\bar{\eta}(s)) : s \in \mathcal{S}\right)$.

**Definition 3.2** (Bellman closedness; (Rowland et al., 2019)). A sketch $\psi_{1:N}$ is **Bellman closed** if there exists an operator $\mathcal{T}_{\psi_{1:N}} : I_{\psi_{1:N}}^{\mathcal{S}} \to I_{\psi_{1:N}}^{\mathcal{S}}$ such that

$$\psi_{1:N}(\mathcal{T}\bar{\eta}) = \mathcal{T}_{\psi_{1:N}} \psi_{1:N}(\bar{\eta}) \quad \text{for all } \bar{\eta} \in \mathscr{P}(\mathbb{R})^{\mathcal{S}}$$

which is closed under a distributional Bellman operator $\mathcal{T} : \mathscr{P}(\mathbb{R})^{\mathcal{S}} \to \mathscr{P}(\mathbb{R})^{\mathcal{S}}$.

Bellman closedness is the property that a sketch are exactly learnable when updates are performed from the infinite-dimensional distribution space to the finite-dimensional embedding space. While classical Bellman equation implies the existence of Bellman operator for expected values, not all statistical functional has such corresponding Bellman operator. Precisely, Rowland et al. (2019) showed that the only finite linear statistical functionals that are Bellman closed are given by the collections of statistical functionals where its linear span is equal to the set of exponential-polynomial functionals. [4]

**Theorem 3.3.** *Quantile functional cannot be Bellman closed under any additional sketch.*

While Rowland et al. (2019) focused on "linear" statistical functionals in defining a sketch (i.e., $\psi(\bar{\eta}) = \mathbb{E}_{Z \sim \bar{\eta}}[h(Z)]$ for some $h$), leaving questions about nonlinear functionals,

---

[4]In discounted setting, a unique solution becomes moments. We've overwritten it for convinience.

we extend this by showing that "nonlinear" statistical functionals, such as maximum or minimum, can also be Bellman closed. Additionally, while their proof implicitly treated quantiles as linear functionals, we provide a technical clarification in Appendix C.1 where we formally demonstrate that no sketch Bellman operator exists for quantiles.

### 3.2. Bellman Unbiasedness

While the intractability caused by infinite-dimensionality was addressed in Bellman closedness, another intractable element which has not yet fully tackled is the *sampling of the next state*. During the implementation, note that the agent does not have access to the transition kernel $\mathbb{P}$. Instead, the agent can only access the empirical transition kernel $\hat{\mathbb{P}}(\cdot|s,a) = \frac{1}{K}\sum_{k=1}^{K} \mathbf{1}\{s'_k = \cdot | s, a\}$ which is derived from $K$ sampled next states. This limitation implies that the operator should be treated as an empirical operator $\hat{\mathcal{T}}_\psi$, rather than $\mathcal{T}_\psi$ (*i.e.*, $\hat{\mathcal{T}}_\psi \psi(\bar{\eta}) := \psi((\mathcal{B}_r)_{\#}[\hat{\mathbb{P}}\bar{\eta}])$). Therefore, we naturally introduce a new notion of *Bellman unbiasedness* to unbiasedly estimate the expected distribution $(\mathcal{B}_r)_{\#}\mathbb{E}_{s' \sim \mathbb{P}(\cdot|s,a)}[\bar{\eta}(s')]$, which is a mixture by transitions, from the sample distribution $(\mathcal{B}_r)_{\#}\bar{\eta}(s')$.

**Definition 3.4** (Bellman unbiasedness). A sketch $\psi(= \psi_{1:N})$ is **Bellman unbiased** if a vector-valued estimator $\phi_\psi = \phi_\psi(\psi(\cdot), \cdots, \psi(\cdot)) : (I_\psi^{\mathcal{S}})^k \to I_\psi^{\mathcal{S}}$ exists where the sketch of expected distribution $(\mathcal{B}_r)_{\#}\mathbb{E}_{s' \sim \mathbb{P}(\cdot|s,a)}[\bar{\eta}(s')]$ can be unbiasedly estimated by $\phi_\psi$ using the $k$ sampled sketches from the sample distribution $(\mathcal{B}_r)_{\#}\bar{\eta}(s')$, i.e.,

$$\mathbb{E}_{s'_i \sim \mathbb{P}}\left[\phi_\psi\left(\underbrace{\psi\Big((\mathcal{B}_r)_{\#}\bar{\eta}(s'_1)\Big), \cdots, \psi\Big((\mathcal{B}_r)_{\#}\bar{\eta}(s'_k)\Big)}_{k \text{ sampled sketches from sample distribution } \hat{\mathcal{T}}_\psi \psi(\bar{\eta}(s))}\right)\right]$$
$$= \psi\Big((\mathcal{B}_r)_{\#}\mathbb{E}_{s' \sim \mathbb{P}(\cdot|s,a)}[\bar{\eta}(s')]\Big).$$

Bellman unbiasedness is another natural definition, similar to Bellman closedness, which takes into account a finite number of samples for the transition. For example, mean-variance sketch is Bellman unbiased as the following unbiased estimator $\phi_{(\mu,\sigma^2)}$ exists for $k$ sample estimates:

$$(\mu, \sigma^2) = \phi_{(\mu,\sigma^2)}\Big((\hat{\mu}_1, \hat{\sigma}_1^2), \cdots, (\hat{\mu}_k, \hat{\sigma}_k^2)\Big)$$
$$= \Big(\frac{1}{k}\sum_{i=1}^{k}\hat{\mu}_i, \ \frac{1}{k}\sum_{i=1}^{k}(\hat{\mu}_i - \frac{1}{k}\sum_{i=1}^{k}\hat{\mu}_i)^2 + \hat{\sigma}_i^2\Big)$$

On the other hand, median functional is not Bellman unbiased since there is no unbiased estimator for median. Then, the following question naturally arises;

*"Which sketches are unbiasmatable under the sketch-based Bellman update?"*

The following lemma answers this question.

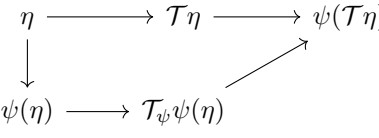

*Figure 3.* Bellman Closedness

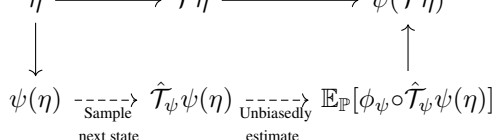

*Figure 4.* Bellman Unbiasedness

*Figure 5.* Illustration of Bellman Closedness and Bellman Unbiasedness. The above path represents an ideal distributional Bellman update. Due to the infinite-dimensionality, the update process should be represented by using a finite-dimensional embedding (sketch) $\psi$. Since the transition kernel $\mathbb{P}$ is unknown, the below path describes that the implementation should sample the next state and update by using $\hat{\mathcal{T}}_\psi$ with the empirical transition kernel $\hat{\mathbb{P}}$. A sketch $\psi$ is Bellman unbiased if $\hat{\mathcal{T}}_\psi \circ \psi$ can unbiasedly estimate $\psi \circ \mathcal{T}$ through some transformation $\phi_\psi$, *i.e.*, $\psi(\mathcal{T}\eta) = \mathbb{E}_{\mathbb{P}}[\phi_\psi \circ \hat{\mathcal{T}}\psi(\eta)]$.

**Lemma 3.5.** *Let $F_{\bar{\eta}}$ be a CDF of the probability distribution $\bar{\eta} \in \mathscr{P}_\psi(\mathbb{R})^{\mathcal{S}}$. Then a sketch is Bellman unbiased if and only if the sketch is homogeneous over $\mathscr{P}_\psi(\mathbb{R})^{\mathcal{S}}$ of degree $k$, i.e., there exists some vector-valued function $h = h(x_1, \cdots, x_k) : \mathcal{X}^k \to \mathbb{R}^N$ such that*

$$\psi(\bar{\eta}) = \int \cdots \int h(x_1, \cdots, x_k) dF_{\bar{\eta}}(x_1) \cdots dF_{\bar{\eta}}(x_k).$$

Lemma 3.5 states that in statistical functional dynamic programming, the unbiasedly estimatable embedding of a distribution can only be structured in the form of functions that are *homogeneous* of finite degree (Halmos, 1946). To illustrate that homogenity defines a broader class than linear functionals, consider the variance as a simple example. Variance is clearly not a linear functional, as it is non-additive. However, it can be written as

$$\text{Var}(\bar{\eta}) = \mathbb{E}_{Z_1 \sim \bar{\eta}}[(Z_1 - \mathbb{E}_{Z_2 \sim \bar{\eta}}[Z_2])^2]$$
$$= \mathbb{E}_{Z_1, Z_2 \sim \bar{\eta}}[Z_1^2 - 2Z_1 Z_2 + Z_2^2] = \mathbb{E}_{Z_1, Z_2 \sim \bar{\eta}}[h(Z_1, Z_2)]$$

which implies the homogenity of degree 2. Taking this concept further and combining it with the results on Bellman closedness, we prove that even when including a nonlinear statistical functional, the only sketch that can be exactly learned and unbiasedly estimated in a finite-dimensional embedding space is the moment sketch.

**Theorem 3.6.** *The only finite sketches that are both Bellman unbiased and closed are given by collections of $\psi_1, \ldots, \psi_N$ where its linear span $\{\sum_{n=0}^{N} \alpha_n \psi_n \mid \alpha_n \in \mathbb{R}, \forall N\}$ is equal to the linear span of the set of exponential polynomial functionals $\{\eta \to \mathbb{E}_{Z \sim \eta}[Z^l \exp(\lambda Z)] \mid l = 0, 1, \ldots, L, \lambda \in \mathbb{R}\}$, where $\psi_0$ is the constant functional equal to $1$.*

Compared to Rowland et al. (2019), we extend beyond linear statistical functionals to include nonlinear statistical functionals, showing the uniqueness of the moment functional. As shown in Figure 1, our theoretical results not only show that high-order central moments such as variance or skewness are exactly learnable and unbiasedly estimatable, but also reveal that other nonlinear statistical functionals like

median or quantiles inevitably involve approximation errors due to biased estimations.

**Necessity of Bellman unbiasedness.** Bellman unbiasedness ensures that updates can be unbiasedly performed when only a finite number of sampled sketches are available. In other words, it guarantees that the sequence of sampled sketches forms a martingale, enabling the construction of confidence regions through concentration inequalities. This property is crucial for establishing provable efficiency in terms of regret minimization.

**Complementary roles of unbiasedness and closedness.** At first glance, Bellman Unbiasedness (BU) may appear to be a stricter subset of Bellman Closedness (BC). However, as illustrated in Figure 1, the relationship is more subtle: for example, the categorical sketch is BU but not BC, whereas functionals like the maximum or minimum are BC but not BU. More precisely, BU guarantees the existence of an unbiased estimator of the ground-truth sketch given a finite number of sampled sketches. In contrast, BC plays a complementary role by ensuring that the update process consistently provide such sketches. If a sketch is BU but not BC–as in the case of the categorical sketch–then the update process cannot continue providing new sampled sketches, making dynamic programming infeasible.

### 3.3. Statistical Functional Bellman Completeness

We consider distributional reinforcement learning with general value function approximation (GVFA). For successful TD learning, GVFA framework for classical RL commonly requires the assumption, *Bellman Completeness*, that after applying Bellman operator, the output lies in the function class $\mathcal{F}$ (Wang et al., 2020; Ayoub et al., 2020; Ishfaq et al., 2021). As a natural extension, our approach receives a tuple of function class $\mathcal{F}^N \subseteq \{f : \mathcal{S} \times \mathcal{A} \to \mathbb{R}^N\}$ as input to represent $N$ moments of distribution. Building on this, we assume that for any $\bar{\eta} : \mathcal{S} \to \mathscr{P}([0, H])$, the sketch of target function lies in the function class $\mathcal{F}^N$.

**Assumption 3.7** (Statistical Functional Bellman Completeness)**.** For any distribution $\bar{\eta} : \mathcal{S} \to \mathscr{P}([0, H])$ and

$h \in [H]$, there exists $f_{\bar{\eta}} \in \mathcal{F}^N$ which satisfies

$$f_{\bar{\eta}}(s, a) = \psi_{1:N}\big((\mathcal{B}_{r_h})_{\#}[\mathbb{P}_h \bar{\eta}](s, a)\big) \quad \forall(s, a) \in \mathcal{S} \times \mathcal{A}$$

**DistBC inevitably leads to linear regret.** In the seminal works, Wang et al. (2023) and Chen et al. (2024) assumed that the function class $\mathcal{H} \subseteq \{\eta : \mathcal{S} \times \mathcal{A} \rightarrow \mathscr{P}([0, H])\}$ follows the *distributional Bellman Completeness* (distBC) assumption (*i.e.,* if $\eta \in \mathcal{H}$ for all $\pi, h \in [H]$, $\mathcal{T}_h^\pi \eta \in \mathcal{H}$). This seems natural, but constructing a finite-dimensional subspace $\mathcal{H}$ that satisfies distBC is quite challenging. Since the distributional Bellman operator is a composition of translation and mixing distributions for the next state, it implies that a function class $\mathcal{H}$ must be closed under translation and mixture. However, when considering the representation of infinite-dimensional distributions using a finite number of representations, it is not trivial that the mixture of distributions can also be represented with the same number of representations. For example, while a Gaussian distribution can be represented using two parameters $(\mu, \sigma^2)$, a mixture of $K$ Gaussians generally requires $2K$ representations.

To avoid the issue of closedness under mixture, both previous studies assumed a discretized reward MDP where all outcomes of the return distribution are able to discretized into an uniform grid of finite points. Unfortunately, the approximation error introduced by the discretization is not negligible when it comes to regret. This is because *model misspecification*, which is the error when the model fails to represent the target, typically leads to linear regret.

**Definition 3.8** (Model Misspecification in distBC). For a given distribution class $\mathcal{H}$ which is the finite-dimensional subspace of the space of all distribution $\mathcal{F}^\infty$, we call $\zeta$ the **misspecification error**

$$\zeta := \inf_{f_{\bar{\eta}} \in \mathcal{H}} \sup_{(s,a) \in \mathcal{S} \times \mathcal{A}} \|f_{\bar{\eta}}(s, a) - (\mathcal{B}_{r_h})_{\#}[\mathbb{P}_h \bar{\eta}](s, a)\|$$

for any $\bar{\eta} : \mathcal{S} \rightarrow \mathscr{P}([0, H])$ and $h \in [H]$.

Note that $\zeta$ is strictly positive unless the function approximator $f_{\bar{\eta}}$ can represent any distribution in the finite-dimensional subspace $\mathcal{H}$ generated by translation and mixture. In a classical linear bandit setting (Zanette et al., 2020), a lower bound with misspecification error $\zeta$ is known to yield linear regret $\Omega(\zeta K)$. Therefore, redefining Bellman Completeness within the infinite-dimensional distribution space is not appropriate, as it either imposes strong constraints on the MDP structure or leads to linear regret. To circumvent model misspecification, we revisit the distributional BC through the statistical functional lens. We propose a novel framework that matches a finite number of statistical functionals to the target, rather than the entire distribution itself.

## 4. `SF-LSVI`: Statistical Functional Least Square Value Iteration

In this section, we propose `SF-LSVI` for distRL framework with general value function approximation. Leveraging the result from Theorem 3.6, we introduce a *moment least square regression*. This allows us to capture a finite set of moment information from the distribution, which can be unbiasedly estimated, thereby leading to the *truncated moment problem* (Shohat & Tamarkin, 1943; Schmüdgen et al., 2017). Unlike previous work (Wang et al., 2023; Chen et al., 2024) that estimates in infinite-dimensional distribution spaces, our method enables to estimate distribution unbiasedly in finite-dimensional embedding spaces without misspecification error. The pseudocode of `SF-LSVI` is described in Appendix B.

**Overview.** At the beginning of episode $k \in [K]$, we maintain all previous samples $\{(s_{h'}^\tau, a_{h'}^\tau, r_{h'}^\tau)\}_{(\tau, h') \in [k-1] \times [H]}$ and initialize a sketch $\psi_{1:N}(\bar{\eta}_{H+1}^k(\cdot)) = \mathbf{0}^N$. For each step $h = H, \ldots, 1$, we compute the normalized sample moments of target distribution $\{(\mathcal{B}_{r_{h'}^\tau})_{\#}\bar{\eta}_{h+1}^k(s_{h'+1}^\tau)\}_{h' \in [H]}$ with the help of binomial theorem,

$$\psi_n\Big((\mathcal{B}_{r_{h'}^\tau})_{\#}\bar{\eta}_h(s_{h'+1}^\tau)\Big) := \frac{\mathbb{E}[(\bar{Z}_{h+1}^k(s_{h'+1}^\tau) + r_{h'}^\tau)^n]}{H^{n-1}}$$

$$= \frac{\sum_{n'=0}^n H^{n'} \psi_{n'}\Big(\bar{\eta}_h(s_{h'+1}^\tau)\Big)(r_{h'}^\tau)^{n-n'}}{H^{n-1}}$$

and iteratively solve the $N$-moment least square regression

$$\tilde{f}_{h,\bar{\eta}}^k \leftarrow \arg\min_{f \in \mathcal{F}} \sum_{\tau=1}^{k-1} \sum_{h'=1}^H \Big( \sum_{n=1}^N f^{(n)}(s_{h'}^\tau, a_{h'}^\tau) - \psi_n\Big((\mathcal{B}_{r_{h'}^\tau})_{\#}\bar{\eta}_{h+1}^k(s_{h'+1}^\tau)\Big)\Big)^2$$

based on the dataset $\mathcal{D}_h^k$ which contains the sketch of temporal target $\psi_{1:N}\Big((\mathcal{B}_{r_{h'}^\tau})_{\#}\bar{\eta}_{h+1}^k(s_{h'+1}^\tau)\Big)$. Then we define $Q_h^k(\cdot, \cdot) = \min\{(\tilde{f}_{h,\bar{\eta}}^k)^{(1)}(\cdot, \cdot) + b_h^k(\cdot, \cdot), H\}$ and choose the greedy policy $\pi_h^k(\cdot)$ with respect to $Q_h^k$. Next, we update all $N$ normalized moments of $Q$-distribution $\psi_{1:N}\Big(\eta_k^h(\cdot, \cdot)\Big)$ and $V$-distribution $\psi_{1:N}\Big(\bar{\eta}_k^h(\cdot)\Big)$. We repeat the procedure until all the $K$ episodes are completed.

## 5. Theoretical Analysis

In this section, we provide the theoretical guarantees for `SF-LSVI` under Assumption 3.7. Applying proof techniques from Wang et al. (2020) and extending the result to a statistical functional lens, we generalize *eluder dimension* (Russo & Van Roy, 2013) to the vector-valued function, which has been widely used in RL literatures (Ayoub et al.,

2020; Wang et al., 2020; Jin et al., 2020) to measure the complexity of learning with the function approximators.

**Definition 5.1** ($\epsilon$-dependent, $\epsilon$-independent, Eluder dimension for vector-valued function). *Let $\epsilon \geq 0$ and $\mathcal{Z} = \{(s_i, a_i)\}_{i=1}^n \subseteq \mathcal{S} \times \mathcal{A}$ be a sequence of state-action pairs.*

- *A state-action pair $(s, a) \in \mathcal{S} \times \mathcal{A}$ is $\boldsymbol{\epsilon}$-dependent on $\mathcal{Z}$ with respect to $\mathcal{F}^N$ if $\|f - g\|_{\mathcal{Z}} \leq \epsilon$ for any vector-valued function $f, g \in \mathcal{F}^N$, then $|f^{(1)}(s, a) - g^{(1)}(s, a)| \leq \epsilon$.*

- *An $(s, a)$ is $\boldsymbol{\epsilon}$-independent on $\mathcal{Z}$ with respect to $\mathcal{F}^N$ if $(s, a)$ is not $\epsilon$-dependent on $\mathcal{Z}$.*

- *The $\boldsymbol{\epsilon}$-eluder dimension $\dim_E(\mathcal{F}^N, \epsilon)$ of a vector-valued function class $\mathcal{F}^N$ is the length of the longest sequence of elements in $\mathcal{S} \times \mathcal{A}$ such that, for some $\epsilon' \geq \epsilon$, every element is $\epsilon'$-independent on its predecessors.*

We assume that the function class $\mathcal{F}^N$ and state-action space $\mathcal{S} \times \mathcal{A}$ have bounded covering numbers.

**Assumption 5.2** (Covering number). *For any $\epsilon > 0$, the following holds:*

- *there exists an $\epsilon$-cover $\mathcal{C}(\mathcal{F}^N, \epsilon) \subseteq \mathcal{F}^N$ with size $|\mathcal{C}(\mathcal{F}^N, \epsilon)| \leq \mathcal{N}(\mathcal{F}^N, \epsilon)$, such that for any $g \in \mathcal{F}^N$, there exists $g' \in \mathcal{C}(\mathcal{F}^N, \epsilon)$ with $\|g - g'\|_{\infty, 1} \leq \epsilon$.*

- *there exists an $\epsilon$-cover $\mathcal{C}(\mathcal{S} \times \mathcal{A}, \epsilon)$ with size $|\mathcal{C}(\mathcal{S} \times \mathcal{A}, \epsilon)| \leq \mathcal{N}(\mathcal{S} \times \mathcal{A}, \epsilon)$, such that for any $(s, a) \in \mathcal{S} \times \mathcal{A}$, there exists $(s', a') \in \mathcal{C}(\mathcal{S} \times \mathcal{A}, \epsilon)$ with $\max_{f \in \mathcal{F}} |f(s, a) - f(s', a')| \leq \epsilon$*

The following two lemmas give confidence bounds on the sum of the $l_2$ norms of all normalized moments.

**Lemma 5.3** (Single Step Optimization Error). *Consider a fixed $k \in [K]$ and a fixed $h \in [H]$. Let $\mathcal{Z}_h^k = \{(s_h^\tau, a_h^\tau)\}_{\tau \in [k-1]}$ and $\mathcal{D}_{h,\bar{\eta}}^k = \left\{\left(s_h^\tau, a_h^\tau, \psi_{1:N}\left((\mathcal{B}_{r_{h'}}^\tau)_{\#}\bar{\eta}(s_{h'+1}^\tau)\right)\right)\right\}_{\tau \in [k-1]}$ for any $\bar{\eta} : \mathcal{S} \to \mathscr{P}([0, H])$. Define $\tilde{f}_{h,\bar{\eta}}^k = \arg\min_{f \in \mathcal{F}^N} \|f\|_{\mathcal{D}_{h,\bar{\eta}}^k}^2$. For any $\bar{\eta}$ and $\delta \in (0, 1)$, there is an event $\mathcal{E}(\bar{\eta}, \delta)$ such that conditioned on $\mathcal{E}(\bar{\eta}, \delta)$, with probability at least $1 - \delta$, for any $\bar{\eta}' : \mathcal{S} \to \mathscr{P}([0, H])$ with $\|\psi_{1:N}(\bar{\eta}') - \psi_{1:N}(\bar{\eta})\|_{\infty, 1} \leq 1/T$, we have*

$$\left\| \tilde{f}_{h,\bar{\eta}'}(\cdot, \cdot) - \psi_{1:N}\left((\mathcal{B}_{r(\cdot, \cdot)})_{\#}[\mathbb{P}\bar{\eta}'](\cdot, \cdot)\right) \right\|_{\mathcal{Z}_h^k}$$
$$\leq c' \left( N^{\frac{1}{2}} H \sqrt{\log(1/\delta) + \log \mathcal{N}(\mathcal{F}^N, 1/T)} \right)$$

*for some constant $c' > 0$.*

Due to the definition of Bellman unbiasedness, we remark that moment sketch has a corresponding vector-valued estimator $\phi_{\psi_{1:N}}$ as an identity and leads to a concentration results as the sampled sketches forms a martingale with respect to the filtration $\mathbb{F}_h^\tau$ induced by the history of $\{(s_h^\tau, a_h^\tau)\}_{\tau \in [k-1]}$ (i.e., $\mathbb{E}\left[\psi_{1:N}\left((\mathcal{B}_{r_h})_{\#}\bar{\eta}(s_h^\tau)\right)\Big|\mathbb{F}_h^\tau\right] = \psi_{1:N}\left((\mathcal{B}_{r_h})_{\#}[\mathbb{P}_h\bar{\eta}](s_h^\tau, a_h^\tau)\right)$).

Another notable aspect in Lemma 5.3 is using normalized moments $\mathbb{E}[Z^n]/H^{n-1}$ instead of moments $\mathbb{E}[Z^n]$, as it reduces the size of the confidence region from $O(H^N)$ to $O(\sqrt{N})$. This adjustment is akin to scaling the optimization function in multi-objective optimization to treat each objective equally, which effectively prevents the model from favoring objectives with larger scales.

**Lemma 5.4** (Confidence Region). *Let $(\mathcal{F}^N)_h^k = \{f \in \mathcal{F}^N | \|f - \tilde{f}_{h,\bar{\eta}}^k\|_{\mathcal{Z}_h^k}^2 \leq \beta(\mathcal{F}^N, \delta)\}$, where*

$$\beta(\mathcal{F}^N, \delta) \geq c' \cdot N H^2 (\log(T/\delta) + \log \mathcal{N}(\mathcal{F}^N, 1/T))$$

*for some constant $c' > 0$. Then with probability at least $1 - \delta/2$, for all $k, h \in [K] \times [H]$, we have*

$$\psi_n\left((\mathcal{B}_{r_h(\cdot, \cdot)})_{\#}[\mathbb{P}_h\bar{\eta}_{h+1}^k](\cdot, \cdot)\right) \in (\mathcal{F}^N)_h^k$$

Lemma 5.4 guarantees that the sequence of moments from the target distribution $\psi_{1:N}\left((\mathcal{B}_{r_h(\cdot, \cdot)})_{\#}[\mathbb{P}_h\bar{\eta}_{h+1}^k](\cdot, \cdot)\right)$ lies in the confidence region $(\mathcal{F}^N)_h^k$ with high probability. Supported by the aforementioned lemma, we can further guarantee that all $Q$-functions are optimistically estimated with high probability and derive our final result.

**Theorem 5.5.** *Under Assumption 3.7, with probability at least $1 - \delta$, `SF-LSVI` achieves a regret bound of*

$$Reg(K) \leq 2H \dim_E(\mathcal{F}^N, 1/T) + 4H\sqrt{KH \log(1/\delta)}$$

Under weaker structural assumptions, we show that `SF-LSVI` enjoys near-optimal regret bound of order $\tilde{O}(d_E H^{\frac{3}{2}}\sqrt{K})$, which is $\sqrt{H}$ better than the state-of-the-art distRL algorithm `V-EST-LSR` (Chen et al., 2024). For the linear MDP setting, we have $d_E = \tilde{O}(d)$ and thus `SF-LSVI` achieves a tight regret bound as $\tilde{O}(\sqrt{d^2 H^3 K})$ which matches a lower bound $\Omega(\sqrt{d^2 H^3 K})$ (Zhou et al., 2021). In our analysis, we highlight two main technical novelties which significantly reduces the degree of regret in distRL framework;

1. We refine previous lemma of Osband et al. (2019) and Wang et al. (2020) to remove the dependency of $\beta(\mathcal{F}^N, 1/\delta)$ (See Appendix D.4), ensuring that regret bound depends only on the pre-defined function class, not on the number of moment extracted.

2. As shown in Table 1, we define the eluder dimension $d_E$ in a finite-dimensional embedding space $\mathcal{F}^N$, while other methods rely on an infinite-dimensional distribution space $\mathcal{H} \subseteq \mathcal{F}^\infty$.

## 6. Conclusions

We describe the sources of approximation error inherent in distribution-based updates and introduce a pivotal concept of Bellman unbiasedness, which enables to exactly learn the information of distribution. We also present a provably efficient online distRL algorithm, `SF-LSVI`, with general value function approximation. Notably, our algorithm achieves a near-optimal regret bound of $\tilde{O}(d_E H^{\frac{3}{2}} \sqrt{K})$, matching the tightest upper bound achieved by non-distributional framework (Zhou et al., 2021; He et al., 2023). One interesting future direction would be to reformulate the definition of regret as discrepencies in moments rather than the expected return, and to show the sample-efficiency of distRL. We hope that our work sheds some light on future research in analyzing the provable efficiency of distRL.

## Acknowledgements

This work is in part supported by the National Research Foundation of Korea (NRF, RS-2024-00451435(40%), RS-2024-00413957(40%)), Institute of Information & communications Technology Planning & Evaluation (IITP, RS-2021-II212068(10%), 2021-0-00180(10%)) grant funded by the Ministry of Science and ICT (MSIT), grant-in-aid of HANHWA SYSTEMS, Samsung Electronics Co., Ltd(IO210202-08370-01), Institute of New Media and Communications(INMAC), and the BK21 FOUR program of the Education and Research Program for Future ICT Samsung Electronics Co., Ltd(IO210202-08370-01), Pioneers, Seoul National University in 2025.

## Impact Statement

This paper presents work whose goal is to advance the field of Machine Learning. There are many potential societal consequences of our work, none which we feel must be specifically highlighted here.

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

# Appendix

## A. Notation

*Table 2.* Table of notation

| Notation | Description |
| --- | --- |
| $\mathcal{S}$ | state space of size $S$ |
| $\mathcal{A}$ | action space of size $A$ |
| $H$ | horizon length of one episode |
| $T$ | number of episodes |
| $r_h(s, a)$ | reward of $(s, a)$ at step $h$ |
| $\mathbb{P}_h(s'\vert s, a)$ | probability transition of $(s, a)$ to $s'$ at step $h$ |
| $\mathbb{H}_h^k$ | history up to step $h$, episode $k$ |
| $N$ | number of statistical functionals |
| $Q_h^\pi(s, a)$ | Q-function of a given policy $\pi$ at step $h$ |
| $V_h^\pi(s)$ | V-function of a given policy $\pi$ at step $h$ |
| $Z_h^\pi(s, a)$ | random variable of $Q$-function |
| $\bar{Z}_h^\pi(s)$ | random variable of $V$-function |
| $\eta_h^\pi(s, a)$ | probability distribution of $Q$-function |
| $\bar{\eta}_h^\pi(s)$ | probability distribution of $V$-function |
| $[\mathbb{P}_h(\cdot)]$ | expectation over transition $[\mathbb{P}_h(\cdot)] = \mathbb{E}_{s'\sim\mathbb{P}_h}(\cdot)$ |
| $(\mathcal{B}_r)_\#$ | pushforward of the distribution through $\mathcal{B}_r(x) := r + x$ |
| $f^{(n)}$ | $n$-th element of $N$-dimensional vector $f$ |
| $\Vert f\Vert_\infty$ | max norm of $f: X \to \mathbb{R}$ defined as $\Vert f\Vert_\infty := \max_{x\in X}\vert f^{(n)}(x)\vert$ |
| $\Vert f\Vert_{\infty,1}$ | $l_1$-norm of max norm of $f: X \to \mathbb{R}$ defined as $\Vert f\Vert_{\infty,1} := \sum_{n=1}^N \max_{x\in X}\vert f^{(n)}(x)\vert$ |
| $\mathcal{F}^N$ | a function class of $N$-dimensional embedding space |
| $\mathcal{Z}$ | a set of state-action pairs $\mathcal{Z} := \{(s_t, a_t)\}_{t=1}^{\vert\mathcal{Z}\vert}$ |
| $\mathcal{D}$ | a dataset $\mathcal{D} := \{(s_t, a_t, [d_t^{(1)}, \cdots, d_t^{(N)}])\}_{t=1}^{\vert\mathcal{D}\vert}$ |
| $\Vert f\Vert_{\mathcal{Z}}^2$ | for $f: \mathcal{S}\times\mathcal{A}\to\mathbb{R}$, define $\Vert f\Vert_{\mathcal{Z}}^2 := \sum_{n=1}^N \sum_{(s,a)\in\mathcal{Z}}(f^{(n)}(s_t, a_t))^2$ |
| $\Vert f\Vert_{\mathcal{D}}^2$ | for $f: \mathcal{S}\times\mathcal{A}\to\mathbb{R}$, define $\Vert f\Vert_{\mathcal{D}}^2 := \sum_{n=1}^N \sum_{t=1}^{\mathcal{D}}(f^{(n)}(s_t, a_t) - d_t^{(n)})^2$ |
| $w^{(n)}(\mathcal{F}^N, s, a)$ | width function of $(s, a)$ defined as $w^{(n)}(\mathcal{F}^N, s, a) := \max_{f,g\in\mathcal{F}^N}\vert f^{(n)}(s, a) - g^{(n)}(s, a)\vert$ |
| $\tilde{f}_{h,\bar{\eta}}^k$ | a solution of moment least squre regression, defined as $\tilde{f}_{h,\bar{\eta}}^k := \arg\min_{f\in\mathcal{F}^N}\Vert f\Vert_{\mathcal{D}_h^k}$ |
| $f_{\bar{\eta}}$ | a target sketch of distribution $\bar{\eta}$, defined as $f_{\bar{\eta}} := \psi_{1:N}((\mathcal{B}_r)_\#[\mathbb{P}_h\bar{\eta}])$ |
| $(\mathcal{F}^N)_h^k$ | a confidence region at step $h$, episode $k$, defined as $(\mathcal{F}^N)_h^k := \{f\in\mathcal{F}^N\vert\ \Vert f - \tilde{f}_{h,\bar{\eta}}^k\Vert_{\mathcal{Z}_h^k}^2 \leq \beta(\mathcal{F}^N, \delta)\}$ |
| $\psi(\bar{\eta})$ | a statistical functional $\mathscr{P}_\psi(\mathbb{R})^{\mathcal{S}} \to \mathbb{R}^S$ |
| $\psi_{1:N}(\bar{\eta})$ | a $N-$collection of statistical functional $\mathscr{P}_{\psi_{1:N}}(\mathbb{R})^{\mathcal{S}} \to \mathbb{R}^{N\times S}$ |
| $\mathscr{P}_{\psi_{1:N}}(\mathbb{R})$ | a domain of sketch $\psi_{1:N}$ |
| $I_{\psi_{1:N}}$ | an image of sketch $\psi_{1:N}$ |
| $\mathcal{T}$ | distributional Bellman operator, defined as $\mathcal{T}\bar{\eta} := (\mathcal{B}_r)_\#[\mathbb{P}\bar{\eta}]$ |
| $\mathcal{T}_\psi$ | sketch Bellman operator w.r.t $\psi$, defined as $\mathcal{T}_\psi\psi(\bar{\eta}) := \psi\left((\mathcal{B}_r)_\#[\mathbb{P}\bar{\eta}]\right)$ |
| $\hat{\mathcal{T}}_\psi$ | empirical sketch Bellman operator w.r.t $\psi$, defined as $\hat{\mathcal{T}}_\psi\psi(\bar{\eta}) := \psi\left((\mathcal{B}_r)_\#[\hat{\mathbb{P}}\bar{\eta}]\right)$ |
| $\mathcal{N}(\mathcal{F}^N, \epsilon)$ | covering number of $\mathcal{F}^N$ w.r.t the $\epsilon-$ball |
| $\dim_E(\mathcal{F}^N, \epsilon)$ | eluder dimension of $\mathcal{F}^N$ w.r.t $\epsilon$ |

# B. Pseudocode of `SF-LSVI` and Technical Remarks

---

**Algorithm 1** Statistical Functional Least Square Value Iteration (`SF-LSVI`($\delta$))

---

**Input:** failure probability $\delta \in (0,1)$ and the number of episodes $K$

1: **for** episode $k = 1, 2, \ldots, K$ **do**
2:     Receive initial state $s_1^k$
3:     Initialize $\psi_{1:N}(\bar{\eta}_{H+1}^k(\cdot)) \leftarrow \mathbf{0}^N$
4:     **for** step $h = H, H-1, \ldots, 1$ **do**
5:         $\mathcal{D}_h^k \leftarrow \left\{ s_{h'}^\tau, a_{h'}^\tau, \psi_{1:N}\left( (\mathcal{B}_{r_{h'}^\tau})_{\#} \bar{\eta}_{h+1}^k(s_{h'+1}^\tau) \right) \right\}_{(\tau, h') \in [k-1] \times [H]}$         // Data collection
6:         $\tilde{f}_{h,\bar{\eta}}^k \leftarrow \arg\min_{f \in \mathcal{F}^N} \|f\|_{\mathcal{D}_h^k}$         // Distribution Estimation
7:         $b_h^k(\cdot, \cdot) \leftarrow w^{(1)}((\mathcal{F}^N)_h^k, \cdot, \cdot)$
8:         $Q_h^k(\cdot, \cdot) \leftarrow \min\{(\tilde{f}_{h,\bar{\eta}}^k)^{(1)}(\cdot, \cdot) + b_h^k(\cdot, \cdot), H\}$
9:         $\pi_h^k(\cdot) = \arg\max_{a \in \mathcal{A}} Q_h^k(\cdot, a), \, V_h^k(\cdot) = Q_h^k(\cdot, \pi_h^k(\cdot))$         // Optimistic planning
10:        $\psi_1\left(\eta_h^k(\cdot, \cdot)\right) \leftarrow Q_h^k(\cdot, \cdot), \, \psi_{2:N}\left(\eta_h^k(\cdot, \cdot)\right) \leftarrow \left( \min\{(\tilde{f}_{h,\bar{\eta}}^k)^{(n)}(\cdot, \cdot), H\} \right)_{n \in [2:N]}$
11:        $\psi_1\left(\bar{\eta}_h^k(\cdot)\right) \leftarrow V_h^k(\cdot), \, \psi_{2:N}\left(\bar{\eta}_h^k(\cdot)\right) \leftarrow \psi_{1:N}\left(\eta_h^k(\cdot, \pi_h^k(\cdot))\right)_{n \in [2:N]}$
12:     **for** $h = 1, 2, \ldots, H$ **do**
13:         Take action $a_h^k \leftarrow \pi_h^k(s_h^k)$
14:         Observe reward $r_h^k(s_h^k, a_h^k)$ and get next state $s_{h+1}^k$.

---

*Remark* B.1. For an optimistic planning, we define the bonus function as the width function $b_h^k(s, a) := w_h^k((\mathcal{F}^N)_h^k, s, a)$ where $(\mathcal{F}^N)_h^k$ denotes a confidence region at step $h$, episode $k$. When $\mathcal{F}$ is a linear function class, the width function can be evaluated by simply computing the maximal distance of weight vector. For a general function class $\mathcal{F}$, computing the width function requires to solve a set-constrained optimization problem, which is known as NP-hard (Dann et al., 2018). However, a width function is computed simply for optimistic exploration, and approximation errors are known to have a small effect on regret (Abbasi-Yadkori et al., 2011).

# C. Related Work and Discussion

## C.1. Technical Clarifications on Linearity Assumption in Existing Results

**Bellman Closedness and Linearity.** Rowland et al. (2019) proved that quantile functional is not Bellman closed by providing a specific counterexample. However, their discussion based on counterexamples can be generalized as it assumes that the sketch Bellman operator for the quantile functional needs to be linear.

They consider an discounted MDP with initial state $s_0$ with single action $a$, which transits to one of two terminal states $s_1, s_2$ with equal probability. Letting no reward at state $s_0$, $\texttt{Unif}([0,1])$ at state $s_1$, and $\texttt{Unif}([1/K, 1+1/K])$ at state $s_2$, the return distribution at state $s_0$ is computed as mixture $\frac{1}{2}\texttt{Unif}([0,\gamma]) + \frac{1}{2}\texttt{Unif}([\gamma/K, \gamma + \gamma/K])$. Then the $\frac{1}{2K}$−quantile at state $s_0$ is $\frac{\gamma}{K}$. They proposed a counterexample where each quantile distribution of state $s_1, s_2$ is represented as $\frac{1}{K}\sum_{k=1}^{K}\delta_{\frac{2k-1}{K}}$ and $\frac{1}{K}\sum_{k=1}^{K}\delta_{\frac{2k+1}{K}}$ respectively, the $\frac{1}{2K}$−quantile of state $s_0$ is $\psi_{q_{2K}}\left(\frac{1}{2K}\sum_{k=1}^{K}\delta_{\frac{\gamma(2k-1)}{K}} + \delta_{\frac{\gamma(2k+1)}{K}}\right) = \frac{3\gamma}{2K}$. However, this example does not consider that the mixture of quantiles is not a quantile of the mixture distribution (i.e., $\psi_q(\lambda\eta_1 + (1-\lambda)\eta_2) \neq \lambda\psi_q(\eta_1) + (1-\lambda)\psi_q(\eta_2)$), due to the nonlinearity of the quantile functional. Therefore, this does not present a valid counterexample to prove that quantile functionals are not Bellman closed.

**Bellman Optimizability and Linearity.** Marthe et al. (2023) proposed the notion of Bellman optimizable statistical functional which redefine the Bellman update by planning with respect to statistical functionals rather than expected returns. They proved that $W_1$-continuous Bellman Optimizable statistical functionals are characterized by exponential utilities $\frac{1}{\lambda}\log\mathbb{E}_{Z\sim\eta}[\exp(\lambda Z)]$. However, their proof requires some technical clarification regarding the assumption that such statistical functionals are linear.

To illustrate, they define a statistical functional $\psi_f$ and consider two probability distributions $\eta_1 = \frac{1}{2}(\delta_0 + \delta_h)$ and $\eta_2 = \delta_{\phi(h)}$ where $\phi(h) = f^{-1}\left(\frac{1}{2}(f(0) + f(h))\right)$. Using the translation property, they lead $\psi_f(\eta_1) = \psi_f(\eta_2)$ to $\frac{1}{2}(f(x) + f(x+h)) = f(x + \phi(h))$ for all $x \in \mathbb{R}$. However, this equality $\psi_f\left(\frac{1}{2}(\delta_x + \delta_{x+h})\right) = \frac{1}{2}(f(x) + f(x+h))$ holds only if $\psi_f$ is linear, which is not necessarily a valid assumption for all statistical functionals.

## C.2. Existence of Nonlinear Bellman Closed Sketch.

The previous two examples may not have considered the possibility that the sketch Bellman operator might not necessarily be linear. However, some statistical functionals are Bellman-closed even if they are nonlinear, so it is open question whether there is a nonlinear sketch Bellman operator that makes the quantile functional Bellman-closed. In this section, we present examples of maximum and minimum functionals that are Bellman-closed, despite being nonlinear.

In a nutshell, consider the maximum of return distribution at state $s_1, s_2$ is $\gamma, \gamma + \gamma/K$ respectively. Beyond linearity, the maximum of return distribution at state $s_0$ can be computed by taking the maximum of these values;

$$\max(\max(\bar{\eta}(s_1)), \max(\bar{\eta}(s_2))) = \max(\gamma, \gamma + \gamma/K) = \gamma + \gamma/K$$

which produces the desired result. This implies the existence of a nonlinear sketch that is Bellman closed. More precisely, by defining $\max_{s'\sim\mathbb{P}(\cdot|s,a)}$ and $\min_{s'\sim\mathbb{P}(\cdot|s,a)}$ as the maximum and minimum of the sampled sketch $\psi\left((\mathcal{B}_r)_\#\bar{\eta}(s')\right)$ with the distribution $\mathbb{P}(\cdot|s,a)$, we can derive the sketch Bellman operator for maximum and minimum functionals as follows;

- $\mathcal{T}_{\psi_{\max}}\left(\psi_{\max}(\bar{\eta}(s))\right) = \max_{s'\sim\mathbb{P}(\cdot|s,a)}\left(\psi_{\max}\left((\mathcal{B}_r)_\#\bar{\eta}(s')\right)\right) = \max_{s'\sim\mathbb{P}(\cdot|s,a)}\left(r + \psi_{\max}\left(\bar{\eta}(s')\right)\right)$

- $\mathcal{T}_{\psi_{\min}}\left(\psi_{\min}(\bar{\eta}(s))\right) = \min_{s'\sim\mathbb{P}(\cdot|s,a)}\left(\psi_{\min}\left((\mathcal{B}_r)_\#\bar{\eta}(s')\right)\right) = \min_{s'\sim\mathbb{P}(\cdot|s,a)}\left(r + \psi_{\min}\left(\bar{\eta}(s')\right)\right).$

## C.3. Non-existence of sketch Bellman operator for quantile functional

In this section, we prove that quantile functional cannot be Bellman closed under any additional sketch. First we introduce the definition of *mixture-consistent*, which is the property that the sketch of a mixture can be computed using only the sketch of the distribution of each component.

**Definition C.1** (mixture-consistent)**.** A sketch $\psi$ is **mixture-consistent** if for any $\nu \in [0, 1]$ and any distributions $\eta_1, \eta_2 \in \mathscr{P}_\psi(\mathbb{R})$, there exists a corresponding function $h_\psi$ such that

$$\psi(\nu\eta_1 + (1 - \nu)\eta_2) = h_\psi\Big(\psi(\eta_1), \psi(\eta_2), \nu\Big).$$

Next, we will provide some examples of determining whether a sketch is mixture-consistent or not.

**Example 1.** Every moment or exponential polynomial functional is mixture-consistent.

*Proof.* For any $n \in [N]$ and $\lambda \in \mathbb{C}$,

$$\mathbb{E}_{Z\sim\nu\eta_1+(1-\nu)\eta_2}[Z^n \exp(\lambda Z)] = \nu\mathbb{E}_{Z\sim\eta_1}[Z^n \exp(\lambda Z)] + (1 - \nu)\mathbb{E}_{Z\sim\eta_2}[Z^n \exp(\lambda Z)].$$

∎

**Example 2.** Variance functional is not mixture-consistent.

*Proof.* Let $\nu = \frac{1}{2}$ and $Z, Y$ be the random variables where $Z \sim \frac{1}{2}\delta_0 + \frac{1}{2}\delta_2$ and $Y \sim \frac{1}{2}\delta_k + \frac{1}{2}\delta_{k+2}$. Then, $\text{Var}(Z) = \text{Var}(Y) = 1$. While RHS is constant for any $k$, LHS is not a constant for any $k$, i.e.,

$$\text{Var}_{X\sim\frac{1}{2}(\frac{1}{2}\delta_0+\frac{1}{2}\delta_2)+\frac{1}{2}(\frac{1}{2}\delta_k+\frac{1}{2}\delta_{k+2})}(X) = \frac{1}{4}(k^2 + 5).$$

∎

While variance functional is not mixture consistent by itself, it can be mixture consistent with another statistical functional, the mean.

**Example 3.** Variance functional is mixture-consistent under mean functional.

*Proof.* Notice that mean functional is mixture-consistent. We need to show that variance functional is mixture-consistent under mean functional.

$$
\begin{aligned}
&\text{Var}_{Z\sim\nu\eta_1+(1-\nu)\eta_2}[Z] \\
&= \mathbb{E}_{Z\sim\nu\eta_1+(1-\nu)\eta_2}[Z^2] - (\mathbb{E}_{Z\sim\nu\eta_1+(1-\nu)\eta_2}[Z])^2 \\
&= \nu\mathbb{E}_{Z\sim\eta_1}[Z^2] + (1 - \nu)\mathbb{E}_{Z\sim\eta_2}[Z^2] - (\nu\mathbb{E}_{Z\sim\eta_1}[Z] + (1 - \nu)\mathbb{E}_{Z\sim\eta_2}[Z])^2 \\
&= \nu(\text{Var}_{Z\sim\eta_1}[Z] + (\mathbb{E}_{Z\sim\eta_1}[Z])^2) + (1 - \nu)(\text{Var}_{Z\sim\eta_2}[Z] + (\mathbb{E}_{Z\sim\eta_2}[Z])^2) \\
&\quad - (\nu\mathbb{E}_{Z\sim\eta_1}[Z] + (1 - \nu)\mathbb{E}_{Z\sim\eta_2}[Z])^2.
\end{aligned}
$$

∎

This means that to determine whether it is mixture-consistent or not, we should check it on a per-sketch basis, rather than on a per-statistical functional basis.

**Example 4.** Maximum and minimum functional are both mixture-consistent.

*Proof.*

$$\max_{Z \sim \nu\eta_1 + (1-\nu)\eta_2}[Z] = \max(\max_{Z \sim \eta_1}[Z], \max_{Z \sim \eta_2}[Z])$$

and

$$\min_{Z \sim \nu\eta_1 + (1-\nu)\eta_2}[Z] = \min(\min_{Z \sim \eta_1}[Z], \min_{Z \sim \eta_2}[Z])$$

∎

Since maximum and minimum functionals are mixture consistent, we can construct a nonlinear sketch bellman operator like the one in section C.2. This is possible because there is a nonlinear function $h_\psi$ that ensures the sketch is closed under mixture.

Before demonstrating that a quantile sketch cannot be mixture consistent under any additional sketch, we will first illustrate with the example of a median functional that is not mixture consistent.

**Example 5.** Median sketch is not mixture-consistent.

*Proof.* Let $\nu = \frac{1}{2}$ and $Z, Y$ be the random variables where $Z \sim 0.2\delta_0 + 0.8\delta_1$ and $Y \sim 0.6\delta_0 + 0.4\delta_k$ for some $0 < k < 1$. Then $\psi_{\mathrm{med}}(Z) = 1$ and $\psi_{\mathrm{med}}(Y) = 0$. However,

$$\mathrm{med}_{X = \frac{Z+Y}{2}}[X] = \psi_{\mathrm{med}}(0.4\delta_0 + 0.2\delta_k + 0.4\delta_1) = k$$

which is dependent in $k$. ∎

**Lemma C.2.** *Quantile sketch cannot be mixture-consistent, under any additional sketch.*

*Proof.* For a given integer $N > 0$ and a quantile level $\alpha \in (0, 1)$, let $\nu = \frac{1}{2}$ and a random variable $Y \sim p_{y_0}\delta_0 + p_{y_1}\delta_{y_1} + \cdots + p_{y_N}\delta_{y_N}$ $(0 < y_1 < \cdots < y_N < 1)$ where $p_{y_0} > \alpha$ so that $\psi_{\alpha-\mathrm{quantile}}[Y] = 0$. Consider another random variable $Z \sim p_{z_0}\delta_0 + p_{z_1}\delta_1$ where $p_{z_0} < \alpha$ so that $\psi_{\alpha-\mathrm{quantile}}[Z] = 1$. Then the $\alpha$−quantile of the mixture $X = \frac{Y+Z}{2}$ is

$$\psi_{\alpha-\mathrm{quantile}}[X] = y_n \text{ where } n = \min\left\{ n \le N \,\bigg|\, \frac{1}{2}\sum_{n'=0}^{n} p_{y_{n'}} + \frac{1}{2}p_{z_0} > \alpha \right\}.$$

Letting $p_{z_0} = 2\alpha - \sum_{n'=0}^{n} p_{y_{n'}}$, we can manipulate $\psi_{\alpha-\mathrm{quantile}}[X]$ to be any value of $y_n$. Hence, $\psi_{\alpha-\mathrm{quantile}}[X]$ is a function of all possible outcomes of $Y$.

If there exists a finite number of statistical functionals which make quantile sketch mixture-consistent, then such sketch would uniquely determine the distribution for any $N$. This results in a contradiction that infinite-dimensional distribution space can be represented by a finite number of statistical functional. ∎

**Lemma C.3.** *If a sketch $\psi$ is Bellman closed, then it is mixture-consistent.*

*Proof.* Consider an MDP where initial state $s_0$ has no reward and transits to two state $s_1, s_2$ with probability $\nu, 1 - \nu$ and reward distribution $\bar{\eta}_1, \bar{\eta}_2$. Since $\psi$ is Bellman closed, $\psi(\bar{\eta}(s_0))$ is a function of $\psi(\bar{\eta}(s_1))$ and $\psi(\bar{\eta}(s_2))$, (i.e., $\psi(\bar{\eta}(s_0))$ $= g_\psi(\psi(\bar{\eta}(s_1)), \psi(\bar{\eta}(s_2)))$ for some $g_\psi$). Since $\psi(\bar{\eta}(s_0)) = \psi(\nu\bar{\eta}(s_1) + (1 - \nu)\bar{\eta}(s_2))$, it implies that $\psi$ is mixture-consistent. ∎

Combining the results of Lemma C.2 and Lemma C.3, we prove that a quantile sketch cannot be Bellman closed, no matter what additional sketches are provided.

# D. Proof

*Theorem* (3.3). Quantile functional cannot be Bellman closed under any additional sketch.

*Proof.* See Lemma C.2 and Lemma C.3. ∎

*Lemma* (3.5). Let $F_{\bar{\eta}}$ be a CDF of the probability distribution $\bar{\eta} \in \mathscr{P}(\mathbb{R})^{\mathcal{S}}$. Then a sketch is Bellman unbiased if and only if the sketch is a homogeneous of degree $k$, i.e., there exists some vector-valued function $h = h(x_1, \cdots, x_k) : \mathcal{X}^k \to \mathbb{R}^N$ such that

$$\psi(\bar{\eta}) = \int \cdots \int h(x_1, \cdots, x_k) dF_{\bar{\eta}}(x_1) \cdots dF_{\bar{\eta}}(x_k).$$

*Proof.* ($\Rightarrow$) Consider an two-stage MDP with a single action $a$, and an initial state $s_0$ which transits to one of terminal state $\{s_1, \cdots, s_K\}$ with transition kernel $\mathbb{P}(\cdot | s_0, a)$. Assume that the reward $r(s_0) = 0$. Then $\bar{\eta}(s_0) = \sum_{k=1}^{K} \mathbb{P}(s_k) \delta_{r(s_k)}$. Note that $s'_1, \cdots, s'_k$ are independent and identically distributed random variable in distribution $\mathbb{P}(\cdot | s, a)$.

$$\mathbb{E}_{s' \sim \mathbb{P}(\cdot | s_0, a)} \left[ \phi_\psi \left( \psi \left( (\mathcal{B}_r)_{\#} \bar{\eta}(s'_1) \right), \cdots, \psi \left( (\mathcal{B}_r)_{\#} \bar{\eta}(s'_k) \right) \right) \right] = \psi_{1:N} \left( (\mathcal{B}_r)_{\#} \mathbb{E}_{s' \sim \mathbb{P}(\cdot | s_0, a)} [\bar{\eta}(s')] \right)$$

$$\implies \mathbb{E}_{s' \sim \mathbb{P}(\cdot | s_0, a)} \left[ \phi_\psi \left( \psi \left( \delta_{r(s'_1)} \right), \cdots, \psi \left( \delta_{r(s'_k)} \right) \right) \right] = \psi \left( \mathbb{E}_{s' \sim \mathbb{P}(\cdot | s_0, a)} [\delta_{r(s')}] \right)$$

$$\implies \mathbb{E}_{s' \sim \mathbb{P}(\cdot | s_0, a)} \left[ \phi_\psi \left( g(s'_1), \cdots, g(s'_k) \right) \right] = \psi \left( \bar{\eta}(s_0) \right)$$

$$\implies \int \cdots \int h(s'_1, \cdots, s'_k) dF_{\bar{\eta}}(s'_1) \cdots dF_{\bar{\eta}}(s'_k) = \psi \left( \bar{\eta}(s_0) \right).$$

($\Leftarrow$)

$$\psi \left( (\mathcal{B}_r)_{\#} \mathbb{E}_{s' \sim \mathbb{P}(\cdot | s, a)} [\bar{\eta}(s')] \right)$$

$$= \int \cdots \int h(x_1, \cdots, x_k) dF_{(\mathcal{B}_r)_{\#} \mathbb{E}_{s' \sim \mathbb{P}(\cdot | s, a)} [\bar{\eta}(s')]}(x_1), \cdots, dF_{(\mathcal{B}_r)_{\#} \mathbb{E}_{s' \sim \mathbb{P}(\cdot | s, a)} [\bar{\eta}(s')]}(x_k)$$

$$= \int \cdots \int h(x_1 + r, \cdots, x_k + r) d \left( \mathbb{E}_{s' \sim \mathbb{P}(\cdot | s, a)} F_{\bar{\eta}(s')}(x_1) \right), \cdots, d \left( \mathbb{E}_{s' \sim \mathbb{P}(\cdot | s, a)} F_{\bar{\eta}(s')}(x_k) \right)$$

$$= \mathbb{E}_{s' \sim \mathbb{P}(\cdot | s, a)} \left[ \int \cdots \int h(x_1 + r, \cdots, x_k + r) dF_{\bar{\eta}(s')}(x_1) \cdots dF_{\bar{\eta}(s')}(x_k) \right]$$

$$= \mathbb{E}_{s' \sim \mathbb{P}(\cdot | s, a)} \left[ \psi \left( (\mathcal{B}_r)_{\#} [\bar{\eta}(s')] \right) \right]$$

∎

*Theorem* (3.6). The only finite statistical functionals that are Bellman unbiased and closed are given by the collections of $\psi_1, \ldots, \psi_N$ where its linear span $\{ \sum_{n=0}^{N} \alpha_n \psi_n | \alpha_n \in \mathbb{R}, \forall N \}$ is equal to the set of exponential polynomial functionals $\{ \eta \to \mathbb{E}_{Z \sim \eta}[Z^l \exp(\lambda Z)] | l = 0, 1, \ldots, L, \lambda \in \mathbb{R} \}$, where $\psi_0$ is the constant functional equal to 1. In discount setting, it is equal to the linear span of the set of moment functionals $\{ \eta \to \mathbb{E}_{Z \sim \eta}[Z^l] | l = 0, 1, \ldots, L \}$ for some $L \leq N$.

*Proof.* Our proof is mainly based on the proof techniques of Rowland et al. (2019) and we describe in an extended form. Since their proof also considers the discounted setting, we will define $\mathcal{B}_{r,\gamma}(x) = r + \gamma x$ for discount factor $\gamma \in [0, 1)$. By assumption of Bellman closedness, $\psi_n \left( (\mathcal{B}_{r,\gamma})_{\#} \bar{\eta}(s') \right)$ will be written as $g(r, \gamma, \psi_{1:N}(\bar{\eta}(s')))$ for some $g$. By assumption of Bellman unbiasedness and Lemma 3.5, both $\psi_{1:N}(\bar{\eta}(s'))$ and $\psi_n \left( (\mathcal{B}_{r,\gamma})_{\#} \bar{\eta}(s') \right)$ are affine as functions of the distribution

$\bar{\eta}(s')$,

$$\psi_{1:N}(\alpha\bar{\eta}_1(s') + (1-\alpha)\bar{\eta}_2(s'))$$
$$= \mathbb{E}_{Z_i \sim \alpha\bar{\eta}_1(s')+(1-\alpha)\bar{\eta}_2(s')}[h_{1:N}(\bar{Z}_1, \cdots, \bar{Z}_k)]$$
$$= \alpha\mathbb{E}_{\bar{Z}_i \sim \bar{\eta}_1(s')}[h_{1:N}(\bar{Z}_1, \cdots, \bar{Z}_k)] + (1-\alpha)\mathbb{E}_{\bar{Z}_i \sim \bar{\eta}_2(s')}[h_{1:N}(\bar{Z}_1, \cdots, \bar{Z}_k)]$$
$$= \alpha\psi_{1:N}(\bar{\eta}_1(s')) + (1-\alpha)\psi_{1:N}(\bar{\eta}_2(s'))$$

and

$$\psi_n\Big((\mathcal{B}_{r,\gamma})_{\#}(\alpha\bar{\eta}_1(s') + (1-\alpha)\bar{\eta}_2(s'))\Big)$$
$$= \mathbb{E}_{Z_i \sim \alpha\bar{\eta}_1(s')+(1-\alpha)\bar{\eta}_2(s')}[h_n(r + \gamma\bar{Z}_1, \cdots, r + \gamma\bar{Z}_k)]$$
$$= \alpha\mathbb{E}_{\bar{Z}_i \sim \bar{\eta}_1(s')}[h_n(r + \gamma\bar{Z}_1, \cdots, r + \gamma\bar{Z}_k)] + (1-\alpha)\mathbb{E}_{\bar{Z}_i \sim \bar{\eta}_2(s')}[h_n(r + \gamma\bar{Z}_1, \cdots, r + \gamma\bar{Z}_k)]$$
$$= \alpha\psi_n\Big((\mathcal{B}_{r,\gamma})_{\#}\bar{\eta}_1(s')\Big) + (1-\alpha)\psi_n\Big((\mathcal{B}_{r,\gamma})_{\#}\bar{\eta}_2(s')\Big)$$

Therefore, $g(r, \gamma, \cdot)$ is also affine on the convex codomain of $\psi_{1:N}$. Thus, we have

$$\mathbb{E}_{\bar{Z}_i \sim \bar{\eta}}[\phi_{\psi_n}(r + \gamma\bar{Z}_1, \cdots, r + \gamma\bar{Z}_k)] = a_0(r, \gamma) + \sum_{n'=1}^{N} a_{n'}(r, \gamma)\mathbb{E}_{\bar{Z}_i \sim \bar{\eta}}[\phi_{\psi_{n'}}(\bar{Z}_1, \cdots, \bar{Z}_k)]$$

for some function $a_{0:N} : \mathbb{R} \times [0,1] \to \mathbb{R}$. By taking $\bar{\eta}(s') = \delta_x$, we obtain

$$\phi_{\psi_n}(r + \gamma x, \cdots, r + \gamma x) = a_0(r, \gamma) + \sum_{n'=1}^{N} a_{n'}(r, \gamma)\phi_{\psi_{n'}}(x, \cdots, x).$$

According to Engert (1970), for any translation invariant finite-dimensional space is spanned by a set of function of the form

$$\{x \mapsto x^l \exp(\lambda_j x) |\, j \in [J], 0 \le l \le L\}$$

for some finite subset $\{\lambda_1, \cdots, \lambda_J\}$ of $\mathbb{C}$. Hence, each function $x \mapsto \phi_{\psi_n}(x, \cdots, x)$ is expressed as linear combination of exponential polynomial functions. In addition, the linear combination of $\phi_{\psi_n}$ should be closed under composition with for any discount factor $\gamma \in [0,1]$, all $\lambda_j$ should be zero. Hence, the linear combination of $\phi_{\psi_1}, \cdots, \phi_{\psi_N}$ must be equal to the span of $\{x \mapsto x^l |\, 0 \le l \le L\}$ for some $L \in \mathbb{N}$.

∎

*Lemma* (5.3). Consider a fixed $k \in [K]$ and a fixed $h \in [H]$. Let $\mathcal{Z}_h^k = \{(s_h^\tau, a_h^\tau)\}_{\tau \in [k-1]}$ and $\mathcal{D}_{h,\bar{\eta}}^k = \Big\{\Big(s_h^\tau, a_h^\tau, \psi_{1:N}\big((\mathcal{B}_{r_{h'}}^\tau)_{\#}\bar{\eta}(s_{h'+1}^\tau)\big)\Big)\Big\}_{\tau \in [k-1]}$ for any $\bar{\eta} : \mathcal{S} \to \mathscr{P}([0,H])$. Define $\tilde{f}_{h,\bar{\eta}}^k = \arg\min_{f \in \mathcal{F}^N} \|f\|_{\mathcal{D}_{h,\bar{\eta}}^k}^2$. For any $\bar{\eta}$ and $\delta \in (0,1)$, there is an event $\mathcal{E}(\bar{\eta}, \delta)$ such that conditioned on $\mathcal{E}(\bar{\eta}, \delta)$, with probability at least $1-\delta$, for any $\bar{\eta}' : \mathcal{S} \to \mathscr{P}([0,H])$ with $\|\psi_{1:N}(\bar{\eta}') - \psi_{1:N}(\bar{\eta})\|_{\infty,1} \le 1/T$ or $\sum_{n=1}^{N} \|\psi_n(\bar{\eta}') - \psi_n(\bar{\eta})\|_\infty \le 1/T$, we have

$$\left\|\tilde{f}_{h,\bar{\eta}'}(\cdot, \cdot) - \psi_{1:N}\Big((\mathcal{B}_{r(\cdot, \cdot)})_{\#}[\mathbb{P}\bar{\eta}'](\cdot, \cdot)\Big)\right\|_{\mathcal{Z}_h^k} \le c'\left(N^{\frac{1}{2}}H\sqrt{\log(1/\delta) + \log\mathcal{N}(\mathcal{F}^N, 1/T)}\right)$$

for some constant $c' > 0$.

*Proof.* Define the sketch of target $f_{\bar{\eta}} : \mathcal{S} \times \mathcal{A} \to \mathbb{R}^N$,

$$f_{\bar{\eta}}(\cdot, \cdot) := \psi_{1:N}\Big((\mathcal{B}_{r(\cdot, \cdot)})_{\#}[\mathbb{P}\bar{\eta}](\cdot, \cdot)\Big)$$

for all $i \in [N]$.

For any $f \in \mathcal{F}$,

$$
\|f\|^2_{\mathcal{D}^k_{h,\bar{\eta}'}} - \|f_{\bar{\eta}'}\|^2_{\mathcal{D}^k_{h,\bar{\eta}'}}
$$

$$
= \sum_{n=1}^{N} \sum_{s^\tau_h, a^\tau_h \in \mathcal{Z}^k_{h,\bar{\eta}'}} \left( f^{(n)}(s^\tau_h, a^\tau_h) - \psi_n\left((\mathcal{B}_{r^\tau_h})_\# \bar{\eta}'(s^\tau_{h+1})\right) \right)^2 - \left( f^{(n)}_{\bar{\eta}'}(s^\tau_h, a^\tau_h) - \psi_n\left((\mathcal{B}_{r^\tau_h})_\# \bar{\eta}'(s^\tau_{h+1})\right) \right)^2
$$

$$
= \sum_{n=1}^{N} \sum_{s^\tau_h, a^\tau_h \in \mathcal{Z}^k_{h,\bar{\eta}'}} (f^{(n)}(s^\tau_h, a^\tau_h) - f^{(n)}_{\bar{\eta}'}(s^\tau_h, a^\tau_h))^2
$$

$$
+ 2(f^{(n)}(s^\tau_h, a^\tau_h) - f^{(n)}_{\bar{\eta}'}(s^\tau_h, a^\tau_h)) \left( f^{(n)}_{\bar{\eta}'}(s^\tau_h, a^\tau_h) - \psi_n\left((\mathcal{B}_{r^\tau_h})_\# \bar{\eta}'(s^\tau_{h+1})\right) \right)
$$

$$
\geq \|f - f_{\bar{\eta}'}\|^2_{\mathcal{Z}^k_h} - 4 \sum_{n=1}^{N} \|f^{(n)}_{\bar{\eta}} - f^{(n)}_{\bar{\eta}'}\|_\infty (H+1)|\mathcal{Z}^k_h|
$$

$$
+ \sum_{n=1}^{N} \sum_{s^\tau_h, a^\tau_h \in \mathcal{Z}^k_{h,\bar{\eta}'}} \Big[ \underbrace{2(f^{(n)}(s^\tau_h, a^\tau_h) - f^{(n)}_{\bar{\eta}}(s^\tau_h, a^\tau_h)) \left( f^{(n)}_{\bar{\eta}}(s^\tau_h, a^\tau_h) - \psi_n\left((\mathcal{B}_{r^\tau_h})_\# \bar{\eta}(s^\tau_{h+1})\right) \right)}_{\chi^\tau_h(f^{(n)})} \Big]
$$

$$
\geq \|f - f_{\bar{\eta}'}\|^2_{\mathcal{Z}^k_h} - 4N(H+1) - \Big| \sum_{n=1}^{N} \sum_{s^\tau_h, a^\tau_h \in \mathcal{Z}^k_{h,\bar{\eta}'}} \chi^\tau_h(f^{(n)}) \Big|.
$$

For the first inequality, we change the second term from $\bar{\eta}'$ to $\bar{\eta}$ which are the $\epsilon$-covers. Notice that $AC - BC' \geq -|AC - BC'| \geq -|(A-B)C| - |(A-B)C'| \geq -2|A-B|\max(C,C')|$.

$$
(f^{(n)}(s^\tau_h, a^\tau_h) - f^{(n)}_{\bar{\eta}'}(s^\tau_h, a^\tau_h)) \left( f^{(n)}_{\bar{\eta}'}(s^\tau_h, a^\tau_h) - \psi_n\left((\mathcal{B}_{r^\tau_h})_\# \bar{\eta}'(s^\tau_{h+1})\right) \right)
$$

$$
- (f^{(n)}(s^\tau_h, a^\tau_h) - f^{(n)}_{\bar{\eta}}(s^\tau_h, a^\tau_h)) \left( f^{(n)}_{\bar{\eta}}(s^\tau_h, a^\tau_h) - \psi_n\left((\mathcal{B}_{r^\tau_h})_\# \bar{\eta}(s^\tau_{h+1})\right) \right)
$$

$$
\geq -2\|f^{(n)}_{\bar{\eta}'}(s^\tau_h, a^\tau_h) - f^{(n)}_{\bar{\eta}}(s^\tau_h, a^\tau_h)\|
$$

$$
\times \max\left( \left| f^{(n)}_{\bar{\eta}'}(s^\tau_h, a^\tau_h) - \psi_n\left((\mathcal{B}_{r^\tau_h})_\# \bar{\eta}'(s^\tau_{h+1})\right) \right|, \left| f^{(n)}_{\bar{\eta}}(s^\tau_h, a^\tau_h) - \psi_n\left((\mathcal{B}_{r^\tau_h})_\# \bar{\eta}(s^\tau_{h+1})\right) \right| \right)
$$

$$
\geq -2\|f^{(n)}_{\bar{\eta}'}(s^\tau_h, a^\tau_h) - f^{(n)}_{\bar{\eta}}(s^\tau_h, a^\tau_h)\|(H+1)
$$

For the second inequality, consider $\bar{\eta}' : \mathcal{S} \to \mathscr{P}([0, H])$ with $\sum_{n=1}^{N} \|\psi_n(\bar{\eta}') - \psi_n(\bar{\eta})\|_\infty \leq 1/T$. We have

$$
\|f^{(n)}_{\bar{\eta}} - f^{(n)}_{\bar{\eta}'}\|_\infty = \max_{s,a} \Big| \sum_{n'=1}^{n} H^{n'}[\psi_{n'}([\mathbb{P}\bar{\eta}](s,a)) - \psi_{n'}([\mathbb{P}\bar{\eta}'](s,a))]r^{n-n'}/H^{n-1} \Big|
$$

$$
\leq \sum_{n'=1}^{n} \max_{s'} \Big| \psi_{n'}(\bar{\eta}(s')) - \psi_{n'}(\bar{\eta}'(s')) \Big|
$$

$$
\leq 1/T.
$$

Defining $\mathbb{F}^k_h$ as the filtration induced by the sequence $\{(s^\tau_{h'}, a^\tau_{h'})\}_{\tau, h' \in [k-1] \times [H]} \cup \{(s^k_1, a^k_1), (s^k_2, a^k_2), \ldots, (s^k_h, a^k_h)\}$, notice

that

$$
\mathbb{E}\Big[ \sum_{n=1}^{N} \chi_h^\tau(f^{(n)}) \Big| \mathbb{F}_h^\tau \Big]
$$

$$
= \sum_{n=1}^{N} 2(f^{(n)}(s_h^\tau, a_h^\tau) - f_{\bar{\eta}}^{(n)}(s_h^\tau, a_h^\tau))(f_{\bar{\eta}}^{(n)}(s_h^\tau, a_h^\tau) - \mathbb{E}\Big[ \psi_n\big( (\mathcal{B}_{r_h^\tau})_\# \bar{\eta}(s_{h+1}^\tau) \big) \Big| \mathbb{F}_h^\tau \Big])
$$

$$
= \sum_{n=1}^{N} 2(f^{(n)}(s_h^\tau, a_h^\tau) - f_{\bar{\eta}}^{(n)}(s_h^\tau, a_h^\tau))(f_{\bar{\eta}}^{(n)}(s_h^\tau, a_h^\tau) - \mathbb{E}_{s_{h+1}^\tau \sim \mathbb{P}_h(\cdot|s_h^\tau, a_h^\tau)}\Big[ \psi_n\big( (\mathcal{B}_{r_h^\tau})_\# \bar{\eta}(s_{h+1}^\tau) \big) \Big])
$$

$$
= \sum_{n=1}^{N} 2(f^{(n)}(s_h^\tau, a_h^\tau) - f_{\bar{\eta}}^{(n)}(s_h^\tau, a_h^\tau))(f_{\bar{\eta}}^{(n)}(s_h^\tau, a_h^\tau) - \psi_n\big( (\mathcal{B}_{r_h^\tau})_\# \mathbb{E}_{s_{h+1}^\tau \sim \mathbb{P}_h(\cdot|s_h^\tau, a_h^\tau)}[\bar{\eta}(s_{h+1}^\tau)] \big))
$$

$$
= 0
$$

and

$$
\Big| \sum_{n=1}^{N} \chi_h^\tau(f^{(n)}) \Big| = \Big| \sum_{n=1}^{N} 2(f^{(n)}(s_h^\tau, a_h^\tau) - f_{\bar{\eta}}^{(n)}(s_h^\tau, a_h^\tau))(f_{\bar{\eta}}^{(n)}(s_h^\tau, a_h^\tau) - \psi_n\big( (\mathcal{B}_{r_h^\tau})_\# \bar{\eta}(s_{h+1}^\tau) \big)) \Big|
$$

$$
\leq \max_{n \in [N]} \Big\{ 2(f_{\bar{\eta}}^{(n)}(s_h^\tau, a_h^\tau) - \psi_n\big( (\mathcal{B}_{r_h^\tau})_\# \bar{\eta}(s_{h+1}^\tau) \big)) \Big\} \sum_{n=1}^{N} \Big| f^{(n)}(s_h^\tau, a_h^\tau) - f_{\bar{\eta}}^{(n)}(s_h^\tau, a_h^\tau) \Big|
$$

$$
\leq 2(H+1) \sum_{n=1}^{N} \Big| f^{(n)}(s_h^\tau, a_h^\tau) - f_{\bar{\eta}}^{(n)}(s_h^\tau, a_h^\tau) \Big|
$$

In third equality, we emphasize that only Bellman unbiased sketch can derive the martingale difference sequence which induce the concentration result. Since every moment functional is commutable with mixing operation, the transformation $\phi_{\psi_n}$ in Definition 3.4 is identity for all $n \in [N]$. Hence, we choose the sketch as moment which already knows $\phi_\psi$.

By Azuma-Hoeffding inequality,

$$
\mathbb{P}\Big[ \Big| \sum_{(\tau,h) \in [k-1] \times [H]} \sum_{n=1}^{N} \chi_h^\tau(f^{(n)}) \Big| \geq \epsilon \Big] \leq 2 \exp\Big( - \frac{\epsilon^2}{2(2(H+1))^2 \sum_{(\tau,h) \in [k-1] \times [H]} \big( \sum_{n=1}^{N} |f^{(n)} - f_{\bar{\eta}}^{(n)}| \big)^2} \Big)
$$

$$
\leq 2 \exp\Big( - \frac{\epsilon^2}{2(2(H+1))^2 \sum_{(\tau,h) \in [k-1] \times [H]} \big( N \sum_{n=1}^{N} |f^{(n)} - f_{\bar{\eta}}^{(n)}|^2 \big)} \Big)
$$

$$
= 2 \exp\Big( - \frac{\epsilon^2}{2N(2(H+1))^2 \|f - f_{\bar{\eta}}\|_{\mathcal{Z}_h^k}^2} \Big)
$$

where the second inequality follows from the Cauchy-Schwartz inequality.

We set

$$
\epsilon = \sqrt{8N(H+1)^2 \|f - f_{\bar{\eta}}\|_{\mathcal{Z}_h^k}^2 \log \Big( \frac{\mathcal{N}(\mathcal{F}^N, 1/T)}{\delta} \Big)}
$$

With union bound for all $f \in \mathcal{C}(\mathcal{F}^N, 1/T)$, with probability at least $1 - \delta$,

$$
\Big| \sum_{(\tau,h) \in [k-1] \times [H]} \sum_{n=1}^{N} \chi_h^\tau(f^{(n)}) \Big| \leq c' N^{\frac{1}{2}} (H+1) \|f - f_{\bar{\eta}}\|_{\mathcal{Z}_h^k} \sqrt{\log \Big( \frac{\mathcal{N}(\mathcal{F}^N, 1/T)}{\delta} \Big)}
$$

for some constant $c' > 0$.

For all $f \in \mathcal{F}^N$, there exists $g \in \mathcal{C}(\mathcal{F}^N, 1/T)$, such that $\|f - g\|_{\infty,1} \leq 1/T$ or $\sum_{n=1}^N \|f^{(n)} - g^{(n)}\|_\infty \leq 1/T$ for all $n \in [N]$,

$$
\left| \sum_{(\tau,h)\in[k-1]\times[H]} \sum_{n=1}^N \chi_h^\tau(f^{(n)}) \right| \leq \left| \sum_{(\tau,h)\in[k-1]\times[H]} \sum_{n=1}^N \chi_h^\tau(g^{(n)}) \right| + 2(H+1)|\mathcal{Z}_h^k| \sum_{n=1}^N \frac{1}{T}
$$

$$
\leq c' N^{\frac{1}{2}}(H+1)\|g - f_{\bar\eta}\|_{\mathcal{Z}_h^k} \sqrt{\log\left(\frac{\mathcal{N}(\mathcal{F}^N, 1/T)}{\delta}\right)} + 2N(H+1)
$$

$$
\leq c' N^{\frac{1}{2}}(H+1)(\|f - f_{\bar\eta}\|_{\mathcal{Z}_h^k} + 1) \sqrt{\log\left(\frac{\mathcal{N}(\mathcal{F}^N, 1/T)}{\delta}\right)} + 2N(H+1)
$$

$$
\leq c' N^{\frac{1}{2}}(H+1)(\|f - f_{\bar\eta'}\|_{\mathcal{Z}_h^k} + 2) \sqrt{\log\left(\frac{\mathcal{N}(\mathcal{F}^N, 1/T)}{\delta}\right)} + 2N(H+1)
$$

where the third inequality follows from,

$$
\|f - g\|_{\mathcal{Z}_h^k}^2 \leq \sum_{n=1}^N \sum_{(\tau,h)\in[k-1]\times[H]} |f^{(n)}(s_h^\tau, a_h^\tau) - g^{(n)}(s_h^\tau, a_h^\tau)|^2
$$

$$
\leq NT\left(\frac{1}{T}\right)^2
$$

$$
\leq 1.
$$

Recall that $\tilde{f}_{h,\eta'}^k = \arg\min_{f\in\mathcal{F}} \|f\|_{\mathcal{D}_{h,\eta'}^k}^2$. We have $\|\tilde{f}_{h,\eta'}^k\|_{\mathcal{D}_{h,\eta'}^k}^2 - \|f_{\bar\eta'}\|_{\mathcal{D}_{h,\eta'}^k}^2 \leq 0$, which implies,

$$
0 \geq \|\tilde{f}_{h,\bar\eta'}^k\|_{\mathcal{D}_{h,\bar\eta'}^k}^2 - \|f_{\bar\eta'}\|_{\mathcal{D}_{h,\bar\eta'}^k}^2
$$

$$
= \|\tilde{f}_{h,\bar\eta'}^k - f_{\bar\eta'}\|_{\mathcal{Z}_h^k}^2
$$

$$
+ 2 \sum_{n=1}^N \sum_{(\tau,h)\in[k-1]\times[H]} \left[ ((\tilde{f}_{h,\bar\eta'}^k)^{(n)}(s_h^\tau, a_h^\tau) - f_{\bar\eta'}^{(n)}(s_h^\tau, a_h^\tau))(f_{\bar\eta'}^{(n)}(s_h^\tau, a_h^\tau) - \psi_n\left((\mathcal{B}_{r_h^\tau})_\# \bar\eta'(s_{h+1}^\tau)\right)) \right]
$$

$$
\geq \|\tilde{f}_{h,\bar\eta'}^k - f_{\bar\eta'}\|_{\mathcal{Z}_h^k}^2 - c' N^{\frac{1}{2}}(H+1)(\|\hat{f}_{h,\bar\eta'}^k - f_{\bar\eta'}\|_{\mathcal{Z}_h^k} + 2) \sqrt{\log(2/\delta) + \log\mathcal{N}(\mathcal{F}^N, 1/T)} - 6N(H+1).
$$

Recall that if $x^2 - 2ax - b \leq 0$ holds for constant $a, b > 0$, then $x \leq a + \sqrt{a^2 + b} \leq c' \cdot a$ for some constant $c' > 0$. Hence,

$$
\|\tilde{f}_{h,\eta'}^k - f_{\bar\eta'}\|_{\mathcal{Z}_h^k} \leq c'(N^{\frac{1}{2}}H\sqrt{\log(1/\delta) + \log\mathcal{N}(\mathcal{F}^N, 1/T)})
$$

for some constant $c' > 0$. ∎

*Lemma* (5.4). Let $(\mathcal{F}^N)_h^k = \{f \in \mathcal{F}^N | \|f - \tilde{f}_{h,\bar\eta}^k\|_{\mathcal{Z}_h^k}^2 \leq \beta(\mathcal{F}^N, \delta)\}$, where

$$
\beta(\mathcal{F}^N, \delta) \geq c' \cdot NH^2(\log(T/\delta) + \log\mathcal{N}(\mathcal{F}^N, 1/T))
$$

for some constant $c' > 0$. Then with probability at least $1 - \delta/2$, for all $k, h \in [K] \times [H]$, we have

$$
\psi_n\left((\mathcal{B}_{r_h(\cdot,\cdot)})_\# [\mathbb{P}_h \bar\eta_{h+1}^k](\cdot,\cdot)\right) \in (\mathcal{F}^N)_h^k
$$

*Proof.* For all $(k, h) \in [K] \times [H]$,

$$
\mathbf{S} := \begin{cases} \left\{ \left(\min\{f^{(1)}(\cdot,\cdot) + b_{h+1}^k(\cdot,\cdot), H\}\right) \middle| f \in \mathcal{C}(\mathcal{F}^N, 1/T) \right\} \cup \{0\} & n = 1 \\ \left\{ \left(\min\{f^{(n)}(\cdot,\cdot), H\}\right) \middle| f \in \mathcal{C}(\mathcal{F}^N, 1/T) \right\} \cup \{0\} & 2 \leq n \leq N \end{cases}
$$

is a $(1/T)$-cover of $\psi_{1:N}(\eta_{h+1}^k(\cdot,\cdot))$ where

$$
\psi_{1:N}(\eta_{h+1}^k(\cdot,\cdot)) = \begin{cases} \min\{(f_{h+1}^k)^{(1)}(\cdot,\cdot) + b_{h+1}^k(\cdot,\cdot), H\} & n = 1 \text{ and } h < H \\ \min\{(f_{h+1}^k)^{(n)}(\cdot,\cdot), H\} & 2 \le n \le N \text{ and } h < H \\ \mathbf{0}^N & h = H \end{cases},
$$

i.e., there exists $\psi_{1:N}(\eta) \in \mathbf{S}$ such that $\|\psi_{1:N}(\eta) - \psi_{1:N}(\eta_{h+1}^k)\|_{\infty,1} \le 1/T$. This implies

$$
\bar{\mathbf{S}} := \left\{ \psi_{1:N}\left( \eta(\cdot, \arg\max_{a \in \mathcal{A}} \psi_1(\eta(\cdot, a))) \right) \mid \psi_{1:N}(\eta) \in \mathbf{S} \right\}
$$

is a $(1/T)$-cover of $\psi_{1:N}(\bar{\eta}_{h+1}^k)$ with $\log(|\bar{\mathbf{S}}|) \le \log \mathcal{N}(\mathcal{F}^N, 1/T)$.

For each $\psi_{1:N}(\bar{\eta}) \in \bar{\mathbf{S}}$, let $\mathcal{E}(\bar{\eta}, \delta/2|\bar{\mathbf{S}}|T)$ be the event defined in Lemma 5.3. By union bound for all $\psi_{1:N}(\bar{\eta}) \in \bar{\mathbf{S}}$, we have $\Pr[\bigcap_{\psi_{1:N}(\bar{\eta}) \in \bar{\mathbf{S}}} \mathcal{E}(\bar{\eta}, \delta/2|\bar{\mathbf{S}}|T)] \ge 1 - \delta/2T$.

Let $\psi_{1:N}(\bar{\eta}) \in \bar{\mathbf{S}}$ such that $\|\psi_{1:N}(\bar{\eta}) - \psi_{1:N}(\bar{\eta}_{h+1}^k)\|_{\infty,1} \le 1/T$. Conditioned on $\bigcap_{s_N(\bar{\eta}) \in \bar{\mathbf{S}}} \mathcal{E}(\bar{\eta}, \delta/2|\bar{\mathbf{S}}|T)$ and by Lemma 5.3, we have

$$
\left\| \tilde{f}_{h,\bar{\eta}}^k(\cdot,\cdot) - \psi_{1:N}\left( (\mathcal{B}_{r_h(\cdot,\cdot)})_\# [\mathbb{P}_h \bar{\eta}_{h+1}^k](\cdot,\cdot) \right) \right\|_{\mathcal{Z}_h^k}^2 \le c'\left( NH^2(\log(T/\delta) + \log \mathcal{N}(\mathcal{F}^N, 1/T)) \right)
$$

for some constant $c' > 0$.

By union bound for all $(k, h) \in [K] \times [H]$, we have $\psi_{1:N}\left( (\mathcal{B}_{r_h(\cdot,\cdot)})_\# [\mathbb{P}_h \bar{\eta}_{h+1}^k](\cdot,\cdot) \right) \in (\mathcal{F}^N)_h^k$ with probability $1 - \delta/2$. ∎

**Lemma D.1.** *Let $Q_h^k(s,a) := \min\{H, \tilde{f}_h^k(s,a) + b_h^k(s,a)\}$ for some bonus function $b_h^k(s,a)$ for all $(s,a) \in \mathcal{S} \times \mathcal{A}$. If $b_h^k(s,a) \ge w^{(1)}((\mathcal{F}^N)_h^k, s, a)$, then with probability at least $1 - \delta/2$,*

$$
Q_h^*(s,a) \le Q_h^k(s,a) \text{ and } V_h^*(s) \le V_h^k(s)
$$

*for all $(k, h) \in [K] \times [H]$, for all $(s,a) \in \mathcal{S} \times \mathcal{A}$.*

*Proof.* We use induction on $h$ from $h = H$ to $1$ to prove the statement. Let $\mathcal{E}$ be the event that for $(k, h) \in [K] \times [H]$, $\psi_{1:N}\left( (\mathcal{B}_{r_h(\cdot,\cdot)})_\# [\mathbb{P}_h \bar{\eta}_{h+1}^k](\cdot,\cdot) \right) \in (\mathcal{F}^N)_h^k$. By Lemma 5.4, $\Pr|\mathcal{E}| \ge 1 - \delta/2$. In the rest of the proof, we condition on $\mathcal{E}$.

When $h = H + 1$, the desired inequality holds as $Q_{H+1}^*(s,a) = V_{H+1}^*(s) = Q_{H+1}^k(s,a) = V_{H+1}^k(s) = 0$. Now, assume that $Q_{h+1}^*(s,a) \le Q_{h+1}^k(s,a)$ and $V_{h+1}^*(s) \le V_{h+1}^k(s)$ for some $h \in [H]$. Then, for all $(s,a) \in \mathcal{S} \times \mathcal{A}$,

$$
\begin{aligned}
Q_h^*(s,a) &= \min\{H, r_h(s,a) + [\mathbb{P}_h V_{h+1}^*](s,a)\} \\
&\le \min\{H, r_h(s,a) + [\mathbb{P}_h V_{h+1}^k](s,a)\} \\
&\le \min\{H, \tilde{f}_h^k(s,a) + w^{(1)}(\mathcal{F}_h^k, s, a)\} \\
&= \min\{H, Q_h^k(s,a) - b_h^k(s,a) + w^{(1)}(\mathcal{F}_h^k, s, a)\} \\
&\le Q_h^k(s,a)
\end{aligned}
$$

∎

**Lemma D.2** (Regret decomposition). *With probability at least $1 - \delta/4$, we have*

$$
Reg(K) \le \sum_{k=1}^{K} \sum_{h=1}^{H} (2b_h^k(s_h^k, a_h^k) + \xi_h^k),
$$

*where $\xi_h^k = [\mathbb{P}_h(V_{h+1}^k - V_{h+1}^{\pi^k})](s_h^k, a_h^k) - (V_{h+1}^k(s_{h+1}^k) - V_{h+1}^{\pi^k}(s_{h+1}^k))$ is a martingale difference sequence with respect to the filtration $\mathbb{F}_h^k$ induced by the history $\mathbb{H}_h^k$.*

*Proof.* We condition on the above event $\mathcal{E}$ in the rest of the proof. For all $(k, h) \in [K] \times [H]$, we have

$$\left\| \tilde{f}_{h,\bar{\eta}}^k(\cdot, \cdot) - \psi_{1:N}\left( (\mathcal{B}_{r_h(\cdot, \cdot)})_{\#}[\mathbb{P}_h \bar{\eta}_{h+1}^k](\cdot, \cdot) \right) \right\|_{\mathcal{Z}_h^k}^2 \leq \beta(\mathcal{F}^N, \delta).$$

Recall that $(\mathcal{F}^N)_h^k = \{f \in \mathcal{F}^N \mid \|f - \tilde{f}_{h,\bar{\eta}}^k\|_{\mathcal{Z}_h^k}^2 \leq \beta(\mathcal{F}^N, \delta)\}$ is the confidence region. Since $\psi_{1:N}\left( (\mathcal{B}_{r_h(\cdot, \cdot)})_{\#}[\mathbb{P}_h \bar{\eta}_{h+1}^k](\cdot, \cdot) \right) \in (\mathcal{F}^N)_h^k$, then by the definition of width function $w^{(1)}(\mathcal{F}_h^k, s, a)$, for $(k, h) \in [K] \times [H]$, we have

$$\begin{aligned}
w^{(1)}(\mathcal{F}_h^k, s, a) &\geq \left| \psi_1\left( (\mathcal{B}_{r_h(s,a)})_{\#}[\mathbb{P}_h \bar{\eta}_{h+1}^k](s, a) \right) - (\tilde{f}_{h,\bar{\eta}}^k)^{(1)}(s, a) \right| \\
&= \left| r_h(s, a) + [\mathbb{P}_h V_{h+1}^k](s, a) - (\tilde{f}_{h,\bar{\eta}}^k)^{(1)}(s, a) \right|.
\end{aligned}$$

Recall that $Q_h^*(\cdot, \cdot) \leq Q_h^k(\cdot, \cdot)$.

$$\begin{aligned}
\text{Reg}(K) &= \sum_{k=1}^K V_1^\star(s_1^k) - V_1^{\pi^k}(s_1^k) \\
&\leq \sum_{k=1}^K V_1^k(s_1^k) - V_1^{\pi^k}(s_1^k) \\
&= \sum_{k=1}^K Q_1^k(s_1^k, a_1^k) - Q_1^{\pi^k}(s_1^k, a_1^k) \\
&= \sum_{k=1}^K Q_1^k(s_1^k, a_1^k) - (r_1(s_1^k, a_1^k) + [\mathbb{P}_1 V_2^k](s_1^k, a_1^k)) + (r_1(s_1^k, a_1^k) + [\mathbb{P}_1 V_2^k](s_1^k, a_1^k)) \\
&\quad - Q_1^{\pi^k}(s_1^k, a_1^k) \\
&\leq \sum_{k=1}^K w^{(1)}((\mathcal{F}^N)_1^k, s_1^k, a_1^k) + b_1^k(s_1^k, a_1^k) + [\mathbb{P}_1(V_2^k - V_2^{\pi^k})](s_1^k, a_1^k) \\
&\leq \sum_{k=1}^K w^{(1)}((\mathcal{F}^N)_1^k, s_1^k, a_1^k) + b_1^k(s_1^k, a_1^k) + (V_2^k(s_2^k) - V_2^{\pi^k}(s_2^k)) + \xi_1^k \\
&\quad \vdots \\
&\leq \sum_{k=1}^K \sum_{h=1}^H (w^{(1)}((\mathcal{F}^N)_h^k, s_h^k, a_h^k) + b_h^k(s_h^k, a_h^k) + \xi_h^k) \\
&\leq \sum_{k=1}^K \sum_{h=1}^H (2 b_h^k(s_h^k, a_h^k) + \xi_h^k)
\end{aligned}$$

∎

It remains to bound $\sum_{k=1}^K \sum_{h=1}^H b_h^k(s_h^k, a_h^k)$, for which we will exploit fact that $\mathcal{F}^N$ has bounded eluder dimension.

**Lemma D.3.** *If $b_h^k(s, a) \geq w^{(1)}((\mathcal{F}^N)_h^k, s, a)$ for all $(s, a) \in \mathcal{S} \times \mathcal{A}$ and $k \in [K]$ where*

$$(\mathcal{F}^N)_h^k = \{f \in \mathcal{F}^N \mid \|f - \tilde{f}_{h,\bar{\eta}}^k\|_{\mathcal{Z}_h^k}^2 \leq \beta(\mathcal{F}^N, \delta)\},$$

*then*

$$\sum_{k=1}^K \sum_{h=1}^H \mathbf{1}\{b_h^k(s_h^k, a_h^k) > \epsilon\} \leq \left( \frac{4\beta(\mathcal{F}^N, \delta)}{\epsilon^2} + 1 \right) dim_E(\mathcal{F}^N, \epsilon)$$

*for some constant $c > 0$.*

*Proof.* We first want to show that for any sequence $\{(s_1, a_1), \ldots, (s_\kappa, a_\kappa)\} \subseteq \mathcal{S} \times \mathcal{A}$, there exists $j \in [\kappa]$ such that $(s_j, a_j)$ is $\epsilon$-dependent on at least $L = \lceil (\kappa - 1)/\dim_E(\mathcal{F}^N, \epsilon) \rceil$ disjoint subsequences in $\{(s_1, a_1), \ldots, (s_{j-1}, a_{j-1})\}$ with respect to $\mathcal{F}^N$. We demonstrate this by using the following procedure. Start with $L$ disjoint subsequences of $\{(s_1, a_1), \ldots, (s_{j-1}, a_{j-1})\}$, $\mathcal{B}_1, \mathcal{B}_2, \ldots, \mathcal{B}_L$, which are initially empty. For each $j$, if $(s_j, a_j)$ is $\epsilon$-dependent on every $\mathcal{B}_1, \ldots, \mathcal{B}_L$, we achieve our goal so we stop the process. Else, we choose $i \in [L]$ such that $(s_j, a_j)$ is $\epsilon$-independent on $\mathcal{B}_i$ and update $\mathcal{B}_i \leftarrow \mathcal{B}_i \cup \{(s_j, a_j)\}$, $j \leftarrow j + 1$. Since every element of $\mathcal{B}_i$ is $\epsilon$-independent on its predecessors, $|\mathcal{B}_i|$ cannot get bigger than $\dim_E(\mathcal{F}^N, \epsilon)$ at any point in this process. Therefore, the process stops at most step $j = L\dim_E(\mathcal{F}^N, \epsilon) + 1 \leq \kappa$.

Now we want to show that if for some $j \in [\kappa]$ such that $b_h^k(s_j, a_j) > \epsilon$, then $(s_j, a_j)$ is $\epsilon$-dependent on at most $4\beta(\mathcal{F}^N, \delta)/\epsilon^2$ disjoint subsequences in $\{(s_1, a_1), \ldots, (s_{j-1}, a_{j-1})\}$ with respect to $\mathcal{F}^N$. If $b_h^k(s_j, a_j) > \epsilon$ and $(s_j, a_j)$ is $\epsilon$-dependent on a subsequence of $\{(s_1', a_1'), \ldots, (s_l', a_l')\} \subseteq \{(s_1, a_1), \ldots, (s_\kappa, a_\kappa)\}$, it implies that there exists $f, g \in \mathcal{F}^N$ with $\|f - \tilde{f}_{h,\bar{\eta}}^k\|_{\mathcal{Z}_h^k}^2 \leq \beta(\mathcal{F}^N, \delta)$ and $\|g - \tilde{f}_{h,\bar{\eta}}^k\|_{\mathcal{Z}_h^k}^2 \leq \beta(\mathcal{F}^N, \delta)$ such that $f^{(1)}(s_t', a_t') - g^{(1)}(s_t', a_t') \geq \epsilon$. By triangle inequality, $\|f - g\|_{\mathcal{Z}_h^k}^2 \leq 4\beta(\mathcal{F}^N, \delta)$. On the other hand, if $(s_j, a_j)$ is $\epsilon$-dependent on $L$ disjoint subsequences in $\{(s_1, a_1), \ldots, (s_\kappa, a_\kappa)\}$, then

$$4\beta(\mathcal{F}^N, \delta) \geq \|f - g\|_{\mathcal{Z}^k}^2 \geq \|f^{(1)} - g^{(1)}\|_{\mathcal{Z}^k}^2 \geq L\epsilon^2$$

resulting in $L \leq 4\beta(\mathcal{F}^N, \delta)/\epsilon^2$. Therefore, we have $(\kappa/\dim_E(\mathcal{F}^N, \epsilon)) - 1 \leq 4\beta(\mathcal{F}^N, \delta)/\epsilon^2$ which results in

$$\kappa \leq \left( \frac{4\beta(\mathcal{F}, \delta)}{\epsilon^2} + 1 \right) \dim_E(\mathcal{F}^N, \epsilon)$$

∎

**Lemma D.4** (Refined version of Lemma 10 in Wang et al. (2020)). *If $b_h^k(s, a) \geq w^{(1)}((\mathcal{F}^N)_h^k, s, a)$ for all $(s, a) \in \mathcal{S} \times \mathcal{A}$ and $k \in [K]$, then*

$$\sum_{k=1}^K \sum_{h=1}^H b_h^k(s_h^k, a_h^k) \leq H\dim_E(\mathcal{F}^N, 1/T).$$

*Proof.* We first sort the sequence $\{b_h^k(s_h^k, a_h^k)\}_{(k,h) \in [K] \times [H]}$ in a decreasing order and denote it by $\{e_1, \ldots, e_T\} (e_1 \geq e_2 \geq \cdots \geq e_T)$. By Lemma D.3, for any constant $M > 0$ and $e_t \geq 1/\sqrt{MT}$, we have

$$t \leq \left( \frac{4\beta(\mathcal{F}^N, \delta)}{Me_t^2} + 1 \right) \dim_E(\mathcal{F}^N, \sqrt{M}e_t) \leq \left( \frac{4\beta(\mathcal{F}^N, \delta)}{Me_t^2} + 1 \right) \dim_E(\mathcal{F}^N, 1/T)$$

which implies

$$e_t \leq \left( \frac{t}{\dim_E(\mathcal{F}^N, 1/T)} - 1 \right)^{-1/2} \sqrt{\frac{4\beta(\mathcal{F}^N, \delta)}{M}},$$

for $t \geq \dim_E(\mathcal{F}^N, 1/T)$. Since we have $e_t \leq H$,

$$\sum_{t=1}^T e_t = \sum_{t=1}^T e_t \mathbf{1}\{e_t < 1/\sqrt{MT}\} + \sum_{t=1}^T e_t \mathbf{1}\{e_t \geq 1/\sqrt{MT}, t < \dim_E(\mathcal{F}^N, 1/T)\}$$

$$+ \sum_{t=1}^T e_t \mathbf{1}\{e_t \geq 1/\sqrt{MT}, t \geq \dim_E(\mathcal{F}^N, 1/T)\}$$

$$\leq \frac{1}{\sqrt{M}} + H\dim_E(\mathcal{F}^N, 1/T) + \sum_{\dim_E(\mathcal{F}^N, 1/T) \leq t \leq T} \left( \frac{t}{\dim_E(\mathcal{F}^N, 1/T)} - 1 \right)^{-1/2} \sqrt{\frac{4\beta(\mathcal{F}^N, \delta)}{M}}$$

$$\leq \frac{1}{\sqrt{M}} + H\dim_E(\mathcal{F}^N, 1/T) + 2\left( \frac{T}{\dim_E(\mathcal{F}^N, 1/T)} - 1 \right)^{1/2} \dim_E(\mathcal{F}^N, 1/T) \sqrt{\frac{4\beta(\mathcal{F}^N, \delta)}{M}}$$

$$= \frac{1}{\sqrt{M}} + H\dim_E(\mathcal{F}^N, 1/T) + \sqrt{16 \cdot \dim_E(\mathcal{F}^N, 1/T) \cdot T \cdot \beta(\mathcal{F}^N, \delta)/M}.$$

Taking $M \to \infty$,

$$\sum_{k=1}^{K} \sum_{h=1}^{H} b_h^k(s_h^k, a_h^k) \le H \dim_E(\mathcal{F}^N, 1/T).$$

$\blacksquare$

*Theorem* (5.5). Under Assumption 3.7, with probability at least $1 - \delta$, `SF-LSVI` achieves a regret bound of

$$\text{Reg}(K) \le 2H \dim_E(\mathcal{F}^N, 1/T) + 4H\sqrt{KH \log(2/\delta)}.$$

*Proof.* Recall that $\xi_h^k = [\mathbb{P}_h(V_{h+1}^k - V_{h+1}^{\pi^k})](s_h^k, a_h^k) - (V_{h+1}^k(s_{h+1}^k) - V_{h+1}^{\pi^k}(s_{h+1}^k))$ is a martingale difference sequence where $\mathbb{E}[\xi_h^k | \mathbb{F}_h^k] = 0$ and $|\xi_h^k| \le 2H$. By Azuma-Hoeffding's inequality, with probability at least $1 - \delta/2$,

$$\sum_{k=1}^{K} \sum_{h=1}^{H} \xi_h^k \le 4H\sqrt{KH \log(2/\delta)}.$$

Conditioning on the above event and Lemma D.4, we have

$$\text{Reg}(K) \le 2 \sum_{k=1}^{K} \sum_{h=1}^{H} b_h^k(s_h^k, a_h^k) + \sum_{k=1}^{K} \sum_{h=1}^{H} \xi_h^k$$
$$\le 2H \dim_E(\mathcal{F}^N, 1/T) + 4H\sqrt{KH \log(2/\delta)}$$

$\blacksquare$

