# OpenReview forum: "Bellman Unbiasedness: Toward Provably Efficient Distributional Reinforcement Learning with General Value Function Approximation"
_ICML.cc/2025/Conference — ICML 2025 poster_

### Official Review · Reviewer_xKcn · 2025-02-26

**Overall Recommendation:** 3

**Summary:**

This paper studies distributional RL, in particular the statistical functional formulation of it. They begin by introducing the concept of Bellman-unbiasedness, obtain results on which equivalent conditions lead to this, and exactly characterize which sets of statistics are both Bellman-unbiased and Bellman-closed, which are the set of the first $N$ moments for any $N>0$. They then introduce an algorithm SF-LSVI which learns these moments, and they prove regret bounds for this algorithm.

**Claims And Evidence:**

I would distill the claims of this paper as (i) introducing and studying the stochastic equivalent of Bellman closedness, and (ii) introducing a distributional RL algorithm SF-LSVI and analyzing its performance.

I believe (i) is well supported, however (ii) is not, for reasons I highlight below.

I think the regret analysis of Section 5 is not what one would want to do in general, and may be answering the wrong question. SF-LSVI is a method for learning a sketch of the first $N$ moments $(s_1,\dots, s_N)$, but the regret bounds in terms of $Reg(K)$ only measure accuracy in terms of the value function, i.e. the accuracy of $s_1$. Theorem 5.5 says nothing about how well $s_2,\dots, s_N$ are learnt, and the bound would hold even if they were learnt arbitrarily poorly. It also appears that additionally learning $s_2,\dots,s_N$ does not improve the regret compared to learning $s_1$ directly. Due to this, I don't believe that the analysis of Section 5 supports the claim that SF-LSVI is an efficient distributional RL algorithm.

**Essential References Not Discussed:**

I don't believe there are any essential references not discussed, at least to my knowledge.

**Experimental Designs Or Analyses:**

N/A (no experiments done)

**Methods And Evaluation Criteria:**

There is no evaluation nor datasets used as this is a work of theoretical nature.

**Other Comments Or Suggestions:**

- There should be a bit more care in the subsection "Distributional Bellman Optimality Equation" of Section 2: $\eta^\star_h$ is *not* uniquely defined, as opposed to the optimal value function.

- In Lemma C.2. the notation $\psi_{q_{\alpha}}$ is used but never defined.

- I believe the authors define a functional to be linear if there exists a function $g$ such that $F(\mu) = \int g d\mu$, however this is never defined in-text. This also overlaps with the definition of homogeneity introduced in Lemma 3.5., I would suggest unifying this notation and cleaning this presentation.

- Section 3 is currently a mix of existing concepts and new concepts/results. I would suggest moving the existing results to a prior section (Section 2 perhaps), so that Section 3 is entirely novel which would make the contribution more clear.

- After Theorem 3.6., it is stated "we extend beyond linear statistical functionals to include nonlinear statistical functionals, showing the uniqueness of the moment functional." This is perhaps misleading since the Bellman unbiased assumption exactly limits to "linear statistical functionals", and this is required in the proof.

- In Example 1 of Appendix Section C, $\lambda$ is overloaded as both the argument of the exponential functional and the mixture coefficient.

**Other Strengths And Weaknesses:**

**Strengths**
- The notion of Bellman unbiasedness is a natural extension of Bellman closedness, and is a valuable step towards understanding and designing sample-based distributional RL algorithms.
- The SF-LSVI algorithm is a nice distributional RL algorithm to study, as in some ways it is the "most similar" to standard RL, and understanding it deeply seems to be a natural step in progress.

**Weaknesses**
- Theorem 3.6. potentially lacks novelty as in light of Lemma 3.5. it essentially reduces to the exact same result of Theorem 4.9. of Rowland (2019) (if I understand correctly).
- The choice of regret analysis is questionable and perhaps not the best, as I discussed above.
- There are a couple of theoretical issues as I discussed above.
- Many parts of the paper can be improved by a careful re-reading to improve the overall flow, at the moment there are a number of minor grammar mistakes and ill-formed sentences.

**Questions For Authors:**

- Rowland et al. (2019) introduce the notion of approximate Bellman closedness for the setting that statistics cannot be learnt exactly (such as quantile RL or categorical RL). What results can be obtained for approximate Bellman unbiasedness?
- Is unbiasedness the "best case" that we'd hope for? What if we have two estimators for a sketch, one which is unbiased and one which is biased but with a lower variance and potentially lower MSE? The bias will likely compound over applications of the operator, but I would imagine if the variance reduction is large enough it could be preferable?

**Relation To Broader Scientific Literature:**

The introduction and analysis of Bellman unbiasedness naturally follows from the analysis of Bellman closedness in literature. The regret analysis is similar to previous regret analysis for distributional RL algorithms in literature, as illustrated in Table 1.

**Theoretical Claims:**

I carefully checked most proofs. I think there are a couple of minor inaccuracies/issues with some results:

- Lemma C.2. states that the quantile sketch is not mixture-consistent for any quantile level $\alpha \in [0,1]$, while Example 4 states that the maximum and minimum functionals are both mixture-consistent. Since these are exactly the quantile sketches for $\alpha=\{0,1\}$ respectively, these results contradict each other.

- There are 2 issues in the proof of Lemma C.2., or at least in my understanding of it. The final step of the proof follows by setting $p_{z_0} = 2\alpha - \sum_{n'=0}^n  p_{z_{y_n'}}$, however I think this could be problematic for two reasons. Firstly we must have $p_{z_0}\geq 0$ as it is a probability measure, however this construction may validate that. Similarly we must also have $p_{z_0}  < \alpha$ as this is an assumption introduced earlier in the proof, but this assumption can also be broken by the given construction.

- Section C.2. claims to prove that the statistical function $\psi_{\text max}$ is Bellman-closed. To do this, they define its corresponding Bellman operator as $T_{\psi_{\text max}(\eta(s))} = \max_{s' \sim P(s,a)} \psi_{\text max} ( {(B_r) \sharp} \eta(s')) $. This is not a valid operator however, as they are applying a nonlinear function to the inside of the statistic $\psi_{\text max}$ on the right-hand side, which is **not** valid for the definition of Bellman-closedness. Instead to do this, writing $u= \psi_{\text max}(\eta(s))$, they should introduce a Bellman operator $T_{\psi_{\text max}}$ such that $T_{\psi_{\text max}(\eta(s))}$ can be written as a function of $u$.

---

> ### Author Rebuttal · Authors · 2025-03-29
>
> We sincerely thank the reviewers for their time and thorough evaluation of our paper. We have organized our responses to your comments below.  **Due to character limits, we have focused on addressing what we considered to be the most important comments. We ask for your understanding that we could not provide responses to all questions.**
>
> ---
> ### 1. Theorem 5.5 says nothing about how well $n$-th moment are learnt.
>
> First, we note that our paper investigates the necessary conditions for achieving provable efficiency and **learning all moments simultaneously in distRL** by using finite-dimensional sketch-based updates.
>
> The reviewer’s concern seems to stem from the observation that standard regret is not well-suited for evaluating the effectiveness of distRL, as it fails to capture discrepancies in higher-order moments (order 2 and above). This may have led to a misunderstanding that SF-LSVI does not achieve moment matching.
> However, Lemmas 5.3 and 5.4 theoretically guarantee that all sketches are learned exactly in finite-dimensional spaces. The reason we use the standard regret in our quantitative evaluation is to follow the conventional regret analysis framework in RL literature.
>
>
> ----
> ### 2. Minor issues on theoretical claims
>
> Thank you for the careful review of these points. It can be assured that the issues you raised do not change the theoretical results.
>
> - (A2-1) In Lemma 3.2, since we derive a contradiction when $p_{y_0}>\alpha$  and $p_{z_0}<\alpha$ , we cannot include cases where $\alpha=0 \text{ or } 1$. Therefore, it is correct to modify it to $\alpha \in (0,1)$, which allows us to draw conclusions consistent with existing max and min functionals.
> - (A2-2) If we understood your second point correctly, we believe the above response addresses it. If not, please let us know and we’d be happy to clarify further.
> - (A2-3) Are you referring to $(\mathcal{B}\_r)$? For the max functional, since $\max_{z \in Z}(Z+r)=\max_{z \in Z}(Z)+r$ (and similarly for the min functional), $T_{\psi_\text{max}}$ and $T_{\psi_\text{min}}$ are valid operators. For clarity, we’ll revise Line 703 as:
>     - $T_{\psi_\text{max}}\Big(\psi_{\text{max}}(\bar{\eta}(s))\Big)= \max_{s' \sim \mathbb{P}(\cdot|s,a)}\Big(r + \psi_{\text{max}}(\bar{\eta}(s'))\Big)$
>
> ---
> ### 3. Theorem 3.6 essentially reduces to the exact same result of [Rowland et al 2019]
>
> Theorem 3.6 extends the results of [Rowland et al. 2019] by encompassing a broader class of statistical functionals.
> Since [Rowland et al. 2019] only focuses on **linear functionals**  (i.e., $s(\mu)=\mathbb{E}_{Z \sim \mu}[h(Z)]$), they cannot demonstrate whether variance is a Bellman closed sketch.
>
> However, since $$\text{Var}(\mu) = \mathbb{E}\_{Z\_1 \sim \mu}[(Z\_1-\mathbb{E}\_{Z\_2 \sim \mu}[Z\_2])^2] = \mathbb{E}\_{Z\_1 ,Z\_2 \sim \mu}[Z\_1 ^2 -2Z\_1 Z\_2+Z\_2^2]=\mathbb{E}\_{Z\_1 ,Z\_2 \sim \mu }[h(Z\_1, Z\_2)],$$ it is a functional that is homogeneous of degree 2. By Lemma 3.5, such functional is Bellman unbiased, and our result, Theorem 3.6 leads to the property that variance is also Bellman closed.
> That is to say, Theorem 3.6 can be summarized as a generalized conclusion that determines whether unbiasedly estimatable statistical functionals, which represent a broader domain, are Bellman closed.
>
> ---
> ### 4. Response to [Other Comments and Suggestions]
>
> (A4-1) While $\eta^{\star}_h$ is generally not uniquely defined, this occurs in cases where $\pi^{\star}$ is not uniquely defined due to the lack of total ordering on distributions in a control case. We will mention this in [Bellemare et al 2017] along with a statement excluding such situations.
>
> (A4-3, 5) We will add text defining linear functionals. As demonstrated earlier with the variance example, homogeneity is a separate concept that does not overlap with linear functionals. Therefore, the statement following Theorem 3.6 is valid.
>
>
> ---
> ### 5. Response to [Questions For Authors]
> While the questions are beyond our current scope, they are insightful, and we would be happy to explore them further. Briefly speaking, we believe that the algorithm cannot achieve the tight regret bound if the bias does not converge to 0 during the learning process.
> However, due to the character limit and the possibility that these points are not central to the reviewer’s evaluation, we have chosen not to focus on them here. If the reviewer’s main concerns have been addressed, we will aim to revisit these topics in the next response round, as space permits.

---

> > ### Comment · Reviewer_xKcn · 2025-04-03
> >
> > I thank the authors for responding to my questions/concerns, and for clearing up my misunderstanding of homogeneous vs linear functionals (please add this distinction to the text!). I'm also happy with their rewriting of the max operator so that readers can clearly tell that it is Bellman-closed, and with the added restriction to Lemma C.2. that $\alpha\in(0,1)$ I think the proof should go through.
> >
> > I acknowledge that Lemmas 5.3 and 5.4 provide guarantees on how well all moments are learnt, but my point still stands that Theorem 5.5. only concerns the first moment, and it feels a bit disappointing that the main result of this paper says nothing about what the algorithm is learning (the first m moments). I think that it wouldn't be a difficult change to modify the notion of regret used and the proof to take this into account, which I would recommend the authors to do, either for this paper or for future work they may do in this area. I also can appreciate that the style of regret result of Theorem 5 has been used in previous analyses of similar distributional RL algorithms, so this is in line with the literature, although for the reasons listed above I don't believe this to be the right choice of analysis.
> >
> > With these comments though I'll raise my score to a weak accept.

---

> > > ### Author Response · Authors · 2025-04-03
> > >
> > > We are glad that our responses have addressed your questions and concerns, and we deeply appreciate your decision to increase the score. For clarity, we will revise the text to include variance as an example to distinguish between the definition of linear functionals and homogeneous.
> > >
> > > ---
> > > ## Difficulties in Redesigning Regret to Reflect Theorem 5.5 and Higher-Order Moments
> > >
> > > We share your concern that, despite proving the consistency of higher-order moments in Lemmas 5.3 and 5.4, Theorem 5.5 only addresses first-order moments.
> > > We attempted to reconstruct a new regret that reflects higher-order moment evaluations but faced technical difficulties in generalizing the proof. While it's possible to take a simple approach of defining regret as the sum of differences between moments, utilizing optimistic estimates like in line 9 of the pseudocode becomes challenging for higher-order moments. Since the optimistic algorithm operates greedily only for first-order moments (i.e., $a^k_h = \arg \max_a Q^k_h(s^k_h,a)$), the relationship $V^k_h(s^k_h)=Q^k_h (s^k_h ,a^k_h)$  holds, but this relationship doesn't hold for higher-order moments, making proof generalization difficult.
> > >
> > > We believe that developing a new regret formulation that circumvents these limitations is necessary. However, we found this to be a quite non-trivial challenge, both conceptually and technically. To maintain clarity and focus in the current paper, we chose not to address these complexities and instead restricted our analysis to conventional regret. We are considering the definition of generalized regret as a future research topic building on this paper.
> > >
> > > ---
> > > ## Response to [Questions For Authors]
> > >
> > > We are happy to address the points we were unable to include in the original rebuttal due to space constraints.
> > >
> > > **(A5-1)**
> > > Defining *Approximate Bellman Unbiasedness* (ABU) is indeed an interesting direction. First, Approximate Bellman Closedness (ABC) is a concept that allows for an average approximation error of sketches up to $\epsilon$.
> > >
> > > $\sup_{(x,a)}\frac{1}{N}\sum_{n=1}^N| \psi_n(\eta_{\pi}(x,a))- \hat{\psi}_n(x,a)| \leq \epsilon$
> > >
> > > Here, $\hat{\psi}_n (x,a)$ represents the value obtained while learning the statistical functional $\psi_n$. Since Bellman closedness is defined in cases where the transition kernel is given, $\hat{\psi}_n$ in ABC refers to the value when the transition kernel is provided.
> > >
> > > On the other hand, Bellman unbiasedness differs from Bellman closedness in that it is defined for cases where unbiased estimation is done through sampling without a transition kernel.
> > > Therefore, when considering ABU, $\hat{\psi}_n$ in the above equation should be interpreted as being estimated by a finite number of samples $\hat{\psi}_n^{(k)}$, and the definition should include the estimation process.
> > >
> > > $\sup_{(x,a)}\min\_{\phi\_{\psi}}\frac{1}{N}\sum\_{n=1}^N \Big| \psi_n(\eta\_{\pi}(x,a))- \mathbb{E}\Big[\phi\_{\psi}\Big(\hat{\psi}\_{n}^{(1)}(x,a), \cdots , \hat{\psi}\_{n}^{(K)}(x,a)\Big)\Big] \Big| \leq \epsilon$
> > >
> > > **(A5-2)** The fundamental issue with having bias lies in the difficulty of analyzing the size of the confidence region. In the case of Bellman unbiased sketches, we can analyze the size of the confidence region through concentration inequality by making the sequence of sketches a martingale. However, when using sketches that are not Bellman unbiased or setting up biased estimators, theoretical development becomes challenging because the applicable concentration results are not clear.
> > >
> > > We expect that to ensure convergence, the estimator must at least be consistent ($\text{Bias}(k) \rightarrow 0$), and to achieve near-optimal regret, it must be asymptotically efficient ($\sqrt{k}\ \text{Bias}(k) \rightarrow 0$). As you mentioned, if some estimators have slow asymptotic convergence rates, We expect they will have proportionally suboptimal regret.

---

### Official Review · Reviewer_iHHe · 2025-03-13

**Overall Recommendation:** 4

**Summary:**

This paper considers learnability and provable efficiency of distributional RL (distRL). The proposed notion of *Bellman unbiasedness* extends *Bellman closedness* in the literature to address the estimation errors stemming from finite samples. They show that moment functionals are the only finite statistical functionals that are both Bellman unbiased and closed. Built on this result, they introduce SF-LSVI for distRL with general function approximation, which enables estimating the distribution unbiasedly in finite-dimensional embedding spaces without misspecification error.

**Claims And Evidence:**

Yes.

**Essential References Not Discussed:**

No.

**Experimental Designs Or Analyses:**

N/A

**Methods And Evaluation Criteria:**

N/A

**Other Comments Or Suggestions:**

No.

**Other Strengths And Weaknesses:**

**Strengths:**

1. The proposed *Bellman unbiasedness* extends *Bellman closeness* to finite sample setting, which is meaningful and important for online/offline RL.
2. The SF-LSVI enables estimating distributions unbiasedly in finite-dimensional embedding spaces, addressing the intractability of implementation in infinite-dimensional space in previous work.

**Weakness:** See Questions.

**Questions For Authors:**

1. Is it possible to show the advantage of SF-LSVI over standard expectation-based learning algorithms? For example, in a cost minimization setting, can it also achieve a small-loss bound as in (Wang et al., 2023)?
2. How would parameter $N$ affect learning and sample complexity?
3. Can you elaborate on the relationship between Bellman unbiasedness and closeness? For example, Figure 1 shows that categorical representation is Bellman unbiased but not closed, but I am unable to find the proof for the argument.
4. In Definition 3.4, $\phi_\psi$ maps $k$ sampled sketches to an estimated one. But I feel like there is an alternative approach where we form an estimation directly on the mixture distributions, i.e., $\\{ (\mathcal{B}\_r)_{\\#} \bar\eta(s_i') \\}\_{i=1}^k$.

**Relation To Broader Scientific Literature:**

N/A

**Theoretical Claims:**

See Questions 3 & 4.

---

> ### Author Rebuttal · Authors · 2025-03-29
>
> We sincerely thank the reviewers for their time and thorough evaluation of our paper. We have organized our responses to your comments below. If any of our responses fail to address the intent of your questions or if you have remaining concerns, please let us know.
>
> ---
> ### 1. The advantage of SF-LSVI over standard expectation-based learning algorithm
>
> SF-LSVI is an algorithm that has the advantage of accurately learning distribution information beyond expectation while maintaining a tight upper bound in terms of standard regret. Therefore, it has the advantage of being able to accurately obtain not only the mean but also various moment information.
>
> However, the standard regret measure currently used to evaluate online RL algorithms cannot measure discrepancy in moments other than the first-order moment (expectation). Due to this inherent limitation of the measure, we do not believe it is suitable for distinguishing between the performance of expectation RL and distRL. As we wrote in the Conclusion section, we believe a generalized definition of regret that also evaluates discrepancies occurring in second and higher-order moments is needed, and we are pursuing this in our follow-up research.
>
> ---
> ### 2. How would parameter N affect learning and sample complexity?
>
> Through Lemma 5.3, since the size of the confidence region increases as $\tilde{O}(\sqrt{N})$, the regret also reflects a factor of $\tilde{O}(\sqrt{N})$. Learning $N$ moments can be viewed as increasing the feature dimension by $N$ times, so space complexity adds a factor of $O(N)$, and according to [Jin et al 2020]'s results, (per-step) computational complexity adds a factor of $O(N^2)$. We will add this explanation to Theorem 5.5 for a deeper understanding of the results.
>
> ---
> ### 3. Relationship between Bellman unbiasedness and closeness
>
> Bellman closedness refers to a property of sketches that allows accurate updating of distribution information when the transition kernel is given. However, in sample-based updates, since the transition kernel is not given, additional sketch properties beyond Bellman closedness are needed. Bellman unbiasedness refers to a complementary property that ensures unbiased learning of sketch updates through finite samples, and through this property, we can guarantee tight upper bounds in regret.
>
> The position of categorical sketch in Figure 1 is based on the results from [Rowland et al 2019]. They showed that categorical sketch is a linear functional (Lemma 3.2 of [Rowland et al 2019]), and by our Lemma 3.5, since all linear functionals are homogeneous over degree 1, categorical sketch is Bellman unbiased.
>
> The fact that categorical sketch is not Bellman closed was proven in Lemma 4.4 of [Rowland et al 2019], so combining these two results leads to the representation in Figure 1. We will clarify this process more explicitly in Figure 1 and Appendix C.
>
> ---
> ### 4. Alternative approach to estimate directly on the mixture distributions
>
> A key distinction between "Bellman unbiasedness" and "Bellman closedness" is that we can learn exact information about the return distribution **without the knowledge of the pre-defined transition kernel**. This means SF-LSVI needs only a finite number of sampled sketches to learn the return distribution unbiasedly, rather than requiring knowledge of the transition kernel.
> As noted in Line 324, this unbiasedness property allows us to transform the learning process into a martingale, construct confidence regions through concentration inequality, and ultimately develop an algorithm that achieves tight upper bounds.
>
> While we could introduce new definitions for sketches (such as max, min) that can be estimated consistently but with bias, constructing confidence regions for non-martingale processes remains largely unexplored without distribution priors. The theoretical analysis of such an approach would likely be extremely challenging.
>
> ---
> ### References
>
> - [Jin et al 2021] : Jin, Chi, Qinghua Liu, and Sobhan Miryoosefi. "Bellman eluder dimension: New rich classes of rl problems, and sample-efficient algorithms." *Advances in neural information processing systems* 34 (2021): 13406-13418.
>
> - [Rowland et al 2019] : Rowland, Mark, et al. "Statistics and samples in distributional reinforcement learning." International Conference on Machine Learning. PMLR, 2019.

---

> > ### Comment · Reviewer_iHHe · 2025-04-02
> >
> > Thank you for the detailed explanation. The response addresses most of my questions. Regarding the relationship between Bellman unbiasedness and closeness, I am curious why Bellman unbiasedness is not a subset of Bellman closeness. As you mentioned in the response, "in sample-based updates ... additional sketch properties beyond Bellman closedness are needed", so it seems to me that Bellman unbiasedness could have been more restrictive.

---

> > > ### Author Response · Authors · 2025-04-02
> > >
> > > We are truly grateful that our responses have helped address your questions and concerns.
> > >
> > > The simplest reason why Bellman unbiasedness (BU) is not a subset of Bellman closedness (BC) is that there exist sketches, like categorical sketches, that are BU but not BC.
> > > To explain the subtle difference, BU means that there exists an unbiased estimator of the ground truth sketch **when given a finite number of sampled sketches.**
> > > Here, BC plays a complementary role by maintaining the condition of **when given a finite number of sampled sketches** during the update process.
> > > Since there is no Categorical Bellman operator that exactly preserves the meaning of categorical sketches during the update process, we cannot obtain a finite number of sampled sketches for the target.
> > > Therefore, while a sketch can be BU without being BC, dynamic programming becomes infeasible in such cases.

---

### Official Review · Reviewer_yros · 2025-03-16

**Overall Recommendation:** 4

**Summary:**

The paper aims to design provably efficient and exactly learnable distributional reinforcement learning algorithm in an online setting, especially under general value function approximation.
For the main findings, they introduce two key properties for statistical functionals:
(1). Bellman Closedness: The sketch (compressed representation) remains consistent under Bellman updates.
(2). Bellman Unbiasedness: The sketch can be unbiasedly estimated using sampled next states.
They find out that only moment functionals (e.g., mean, variance, higher-order moments) satisfy both properties and prove that quantile-based functionals (like those used in QR-DQN) are neither closed nor unbiased.
This work proposes Statistical Functional Least Squares Value Iteration (SF-LSVI) that focuses on matching a finite number of moments of the distribution instead of fitting the full distribution using a learnable and unbiased moment-based Bellman update.
Rather than estimating the full return distribution (which is infinite-dimensional), the paper learns a finite-dimensional sketch composed of moment functionals. These sketches are provably closed under Bellman updates and can be unbiasedly estimated from samples, making them ideal for regret analysis and online learning.

**Claims And Evidence:**

Yes, the claims made in the submission are supported by clear and convincing evidence.

**Essential References Not Discussed:**

Related works are well discussed.

**Experimental Designs Or Analyses:**

The main contribution of this work is theoretical, aimed at addressing foundational issues in distributional RL. Instead of providing empirical experiments, the paper provides non-trivial regret bounds and tight complexity analysis, which is already valuable.
The proposed method (SF-LSVI) fills a known theoretical gap: regret-optimal DistRL under general function approximation.

**Methods And Evaluation Criteria:**

Yes, the proposed methods and evaluation criteria make sense and are well-aligned with the problem the paper addresses.

**Other Comments Or Suggestions:**

Empirical validation can be considered, so that we can compare proposed method with quantile based approaches such as QRDQN and IQN in different tasks.

**Other Strengths And Weaknesses:**

The paper demonstrates strong originality by introducing the novel concept of "Bellman unbiasedness" and showing that moment functionals are uniquely suited for provably efficient distributional RL.
Its theoretical contributions are significant, addressing long-standing challenges in DistRL such as the intractability of full-distribution learning and model misspecification.
The work clearly builds on and advances the literature in a well-structured and technically rigorous way.
However, one notable weakness is the lack of empirical validation to support the theoretical findings.
Overall, the paper is clear and well-motivated, with impactful insights for the theory of distributional reinforcement learning.

**Questions For Authors:**

I am curious about the potential applicability of this approach in deep reinforcement learning scenarios.
Although quantile-based methods have known theoretical limitations, algorithms such as QR-DQN and IQN have demonstrated strong empirical performance across a range of risk-sensitive tasks.
It would be interesting to explore whether representations based on moment functionals can be effectively integrated with neural networks, and whether such integration could lead to performance improvements over existing quantile-based approaches like QR-DQN.

**Relation To Broader Scientific Literature:**

This paper extends prior work on distributional reinforcement learning and general value function approximation by building on Bellman closedness (Rowland et al., 2019) and eluder dimension-based regret analysis (Wang et al., 2020).
Unlike previous approaches relying on quantile or full-distribution representations, the authors show that only moment functionals satisfy both Bellman closedness and unbiasedness. They redefine Bellman completeness through a moment-based lens, addressing model misspecification issues found in works like Chen et al. (2024).
This work leads to SF-LSVI, a distributional RL algorithm with provable regret guarantees and a strengthened theoretical foundation.

**Theoretical Claims:**

(1) Theorem 3.3 shows quantile functional cannot be Bellman closed. It aligns with existing theory.
(2) Definition of Bellman Unbiasedness and Lemma 3.5 claim: only statistical functionals that are homogeneous of finite degree can be unbiasedly estimated under sketch-based Bellman updates.
The logic follows known properties of estimators; similar arguments are used in kernel mean embedding and U-statistics.
(3) Theorem 3.6 shows only moment functionals are both Bellman closed and unbiased, it's consistent with Rowland et al. (2019) and standard function approximation theory.
(4) Lemma 5.4 Confidence bound uses martingale concentration inequalities and normalized moment scaling.
(5) SF-LSVI achieves a regret bound expressed in Theorem 5.5 based on Lemma 5.4, Eluder dimension theory and Martingale-based sketch estimation. The derivation path is well aligned with previous work, with careful adjustments for statistical functional setting.

---

> ### Author Rebuttal · Authors · 2025-03-29
>
> We sincerely thank the reviewers for their time and thorough evaluation of our paper. We have organized our responses to your comments below. If any of our responses fail to address the intent of your questions or if you have remaining concerns, please let us know.
>
> ----
> ### 1. Lack of empirical validation
>
> While we acknowledge the value of experimental results, our primary aim is to establish theoretical connections between distributional RL and General Value Function Approximation (GVFA). Our main contribution lies in developing theoretical foundations and deepening understanding of these fields. We note that many GVFA papers with similar theoretical objectives [Jin et al 2021, Li et al 2024] likewise focus on theoretical advances without experimental validation.
>
>
>
> ---
> ### 2. Potential applicability in deep reinforcement learning scenarios
>
> Thank you for your interest in deep RL applications. Our SF-LSVI algorithm, which learns distributions via moment matching, can be related to MMDQN [Nguyen et al., 2021] when adapted to deep RL. Notably, MMDQN has already demonstrated superior performance compared to C51, QRDQN, and IQN.
>
> While MMDQN uses particle-based moment matching, SF-LSVI explicitly constructs and updates predefined moment functionals. Since it is well known that, in truncated moment problems, 10-20 moments typically suffice for reconstructing distributions, we could learn the sketches of distribution (moments) more efficiently than existing distRL methods that require 50-200 statistical functionals.
>
> ---
> ### References
>
> - [Jin et al 2021] : Jin, Chi, Qinghua Liu, and Sobhan Miryoosefi. "Bellman eluder dimension: New rich classes of rl problems, and sample-efficient algorithms." *Advances in neural information processing systems* 34 (2021): 13406-13418.
> - [Li et al 2024] : Li, Yunfan, and Lin Yang. "On the model-misspecification in reinforcement learning." *International Conference on Artificial Intelligence and Statistics*. PMLR, 2024.
> - [Nguyen et al 2021] : Nguyen-Tang, Thanh, Sunil Gupta, and Svetha Venkatesh. "Distributional reinforcement learning via moment matching." *Proceedings of the AAAI Conference on Artificial Intelligence*. Vol. 35. No. 10. 2021.

---

### Official Review · Reviewer_fMkS · 2025-03-16

**Overall Recommendation:** 3

**Summary:**

The paper proposes a distributional RL algorithm in the finite horizon episodic MDP setting. They propose bellman unbiasedness, a notion complementary to bellman closeness in prior work. They analyze the regret bound of the algorithm and compare against prior work in the space, showcasing theoretical improvements.

**Claims And Evidence:**

The theoretical claims made in the paper are fairly clear and backed up by proof.

**Essential References Not Discussed:**

NA

**Experimental Designs Or Analyses:**

No empirical designs.

**Methods And Evaluation Criteria:**

There is no empirical evaluation of the theoretical results in this work, which arguably is a place for improving the current paper.

**Other Comments Or Suggestions:**

NA

**Other Strengths And Weaknesses:**

The paper is fairly solidly grounded in theoretical discussions and it has made solid theoretical contributions connecting the learning efficiency of distributional RL as measured in regret. I think the paper can use a bit more improvement in presenting concrete examples of statistical functionals to make the results more accessible to algorithmic minded readers and add a section for empirical evaluation.

**Questions For Authors:**

=== *concrete examples of statistical functionals* ===

I think the paper will be made more accessible to readers less versed in theoretical discussions, to provide a bit more concrete examples on statistical functionals that fall into different categories. For example, in the appendix maybe discuss why max and min functionals are bellman closed (it is easier to see why they cannot be estimated in an unbiased way) and why categorical functionals are not bellman closed.

=== *bellman unbiasedness vs. closeness* ===

I would love to understand better the contribution this work makes in relation to results in Rowland et al 2019. Rowland showed that the only finite bellman closed statistical functions are spanned by finite moments, where as in here the illustration shows that min and max functionals are also bellman closed?

One implication of bellman closeness is that the statistical functionals can be learned via a recursive bellman backup, does that mean in principle max and min statistical functionals can also be obtained by bellman backup and computed as a fixed point for dynamic programming (though not computable in finite samples due to the lack of unbiasedness).

=== *translating results to discount case* ===

The discussions are limited to finite horizon episodic MDP - I wonder what happens if we consider infinite horizon discounted MDP with discount $\gamma$, how should we translate the regret bound in table 1 as a function of $\gamma$ or is this feasible at all?

=== *empirical validation* ===

I think the paper will benefit greatly from even a simple empirical validation of the results - simulating the regret bound as you would compute in theory, with a tabular mdp environment and see how theoretical insights might be validated. This will be valuable to more empirically minded readers and make a better case for the theoretical results in this work.

**Relation To Broader Scientific Literature:**

The paper is generally related to distributional RL and theoretical RL on regret bound for learning efficiency.

**Theoretical Claims:**

I have skimmed through certain theoretical arguments in the paper and they generally are sensible to me.

---

> ### Author Rebuttal · Authors · 2025-03-29
>
> We sincerely thank you for your time and effort in reviewing our paper. We have organized our responses to your comments below. If any of our reconstructed responses miss the intent of your questions or if there are remaining concerns, please let us know so we can address them.
>
> -------
> ### 1. Concrete examples of statistical functionals
>
> In our paper, we indicated in Figure 1 that max and min are Bellman closed but not unbiased statistical functionals, and the proof for this is provided in Appendix C.2. To make it easier to follow, we will add a note in the main text stating that 'the proof is included in the appendix.' For the categorical sketch, since  Lemma 4.4 of [Rowland et al 2019] proves that it is not Bellman closed, we did not include a separate proof. We will update Figure 1 in the main text to include a reference to their proven result.
>
>
> -----
> ### 2. Bellman unbiasedness vs Closedness
>
> First, Theorem 4.3 from [Rowland et al 2019] states:
>
> > The only finite sets of statistics of the form $s(\mu)=\mathbb{E}_{Z\sim\mu}[h(Z)]$ that are Bellman closed are ...
> >
>
> In other words, they provide theoretical results for sets of "linear" statistical functionals that satisfy Bellman closedness. Since max and min are non-linear statistical functionals that fall outside the scope of their theory, we cannot determine whether they are Bellman closed.
>
> Similarly, since variance is nonlinear, we cannot determine its Bellman closedness using their results. However, since we already know that first and second moments are Bellman closed, we can naturally derive that variance is Bellman closed. Therefore, this indicates that their theory does not sufficiently cover various commonly used statistical functionals. Keeping this in mind and comparing with Rowland's theory, our theory can be interpreted as a generalized result that is helpful to test Bellman closedness for a broader category of unbiasedly estimatable statistical functionals.
>
> Although the second question falls outside the scope of our paper, it raises important points to address by breaking it into two parts.
>
> > "For a given transition kernel, does a nonlinear Bellman closed sketch always have a fixed point?"
>
> Since Lemma 3 in [Bellemare et al 2017] proves that the distributional update is a contraction, we can see that statistical functionals with bounded values also converge to a fixed point.
>
> > "In sample-based updates without a given transition kernel, does a nonlinear Bellman closed sketch always have a fixed point?"
>
> Our paper proves convergence for Bellman unbiased sketches in a scenario with "only finite sampling allowed without a given transition kernel," but does not examine convergence for other Bellman closed sketches. Since max and min are Bellman closed but nonlinear, we cannot use existing linearity-based contraction proofs. A separate proof approach would be needed, making this an interesting direction for future work.
>
> -----
> ### 3. Translating results to discount case
>
> While regret analysis for the infinite horizon discounted case falls outside the scope of our current paper, we believe it presents an interesting problem. In our results, when performing $N$ sketch-based updates, there is no additional cost beyond the confidence region increasing by a factor of $\sqrt{N}$, so we expect that in the discount case, there would also be an additional factor of $\sqrt{N}$ involved.
>
> ----
> ### 4. Lack of empirical validation
>
> While we acknowledge the importance of experimental results, our paper aims to theoretically connect two fields - distRL and General Value Function Approximation (GVFA) - so our main contribution lies in establishing theoretical foundations and broadening understanding. We kindly request that you consider this in your evaluation, as many GVFA papers with similar objectives [Jin et al 2021, Li et al 2024] also do not necessarily include experimental validation.
>
>
>
> ----
> ### References
>
> - [Rowland et al 2019] : Rowland, Mark, et al. "Statistics and samples in distributional reinforcement learning." *International Conference on Machine Learning*. PMLR, 2019.
> - [Bellemare et al 2017] : Bellemare, Marc G., Will Dabney, and Rémi Munos. "A distributional perspective on reinforcement learning." *International conference on machine learning*. PMLR, 2017.
> - [Jin et al 2021] : Jin, Chi, Qinghua Liu, and Sobhan Miryoosefi. "Bellman eluder dimension: New rich classes of rl problems, and sample-efficient algorithms." *Advances in neural information processing systems* 34 (2021): 13406-13418.
> - [Li et al 2024 ] : Li, Yunfan, and Lin Yang. "On the model-misspecification in reinforcement learning." *International Conference on Artificial Intelligence and Statistics*. PMLR, 2024.

---

### Decision · Program_Chairs · 2025-05-01

**Decision:**

Accept (poster)

**Comment:**

The paper got overall positive reviews and the reviewers highlight in particular the novelty of the paper. I therefore recommend to accept the paper. There are a range of suggestions by the reviewers to improve the paper and I hope that the authors will consider these suggestions for improving the final version of the paper.